# LISTEN, THINK, AND UNDERSTAND

**Yuan Gong**[1✉]**, Hongyin Luo**[1]**, Alexander H. Liu**[1]**, Leonid Karlinsky**[2]**, James Glass**[1]
MIT CSAIL[1]     MIT-IBM Watson AI Lab[2]
{yuangong,hyluo,alexhliu,glass}@mit.edu, leonidka@ibm.com

## ABSTRACT

The ability of artificial intelligence (AI) systems to perceive and comprehend audio signals is crucial for many applications. Although significant progress has been made in this area since the development of AudioSet, most existing models are designed to map audio inputs to pre-defined, discrete sound label sets. In contrast, humans possess the ability to not only classify sounds into general categories, but also to *listen* to the finer details of the sounds, *explain* the reason for the predictions, *think* about what the sound infers, and *understand* the scene and what action needs to be taken, if any. Such capabilities beyond perception are not yet present in existing audio models. On the other hand, modern large language models (LLMs) exhibit emerging reasoning ability but they lack audio perception capabilities. Therefore, we ask the question: can we build a model that has both audio perception *and* reasoning ability?

In this paper, we propose a new audio foundation model, called LTU (Listen, Think, and Understand). To train LTU, we created a new OpenAQA-5M dataset consisting of 1.9 million closed-ended and 3.7 million open-ended, diverse (audio, question, answer) tuples, and have used an autoregressive training framework with a perception-to-understanding curriculum. LTU demonstrates strong performance and generalization ability on conventional audio tasks such as classification and captioning. More importantly, it exhibits emerging audio reasoning and comprehension abilities that are absent in existing audio models. To the best of our knowledge, LTU is the first multimodal large language model that focuses on general audio (rather than just speech) understanding.

## 1 INTRODUCTION

As we go about our daily lives, we are surrounded by complex mixtures of audio signals. Our cognitive abilities enable us not only to perceive and identify these sounds but also to comprehend their implicit meaning. For instance, upon hearing a clock chime six times, we probably assume it's 6 o'clock; when hearing a train whistle we might infer the arrival or departure of a train; when encountering unfamiliar animal sounds in the wilderness, we possess the ability to assess potential danger by discerning certain audio characteristics. Additionally, we can also grasp the emotional atmosphere conveyed by audio cues. We anticipate that future artificial intelligence systems will have similar abilities to perceive and comprehend an auditory environment.

Thanks to the release of the large-scale audio corpus AudioSet (Gemmeke et al., 2017), significant advances have been achieved in general audio event recognition over the past five years. Notably, the mean Average Precision (mAP) for audio tagging has improved significantly, increasing from 31.4 (Gemmeke et al., 2017) to 47.3 (Huang et al., 2022), demonstrating the increasingly strong capabilities of deep learning models to perceive sounds. However, models trained with discrete sound label sets (e.g., the 527-category AudioSet), while being strong in *perception*, possess very limited *reasoning* and *understanding* capabilities. For example, the model may recognize a clock chime 6 times, but not know that it indicates a time of 6 o'clock. On the other hand, we find that modern large language models (LLMs), such as GPT (Brown et al., 2020; OpenAI, 2023) and LLaMA (Touvron et al., 2023) possess strong audio knowledge and reasoning abilities without any in-domain audio training or fine-tuning, e.g., ChatGPT outputs "dog bark is short, repetitive, and sharp woof or bark sound" when we ask it the question "what does a dog bark sound like?" However, even the most advanced LLMs are not yet able to perceive general audio events.

---

Code, dataset, and pretrained models are available at https://github.com/yuangongnd/ltu.

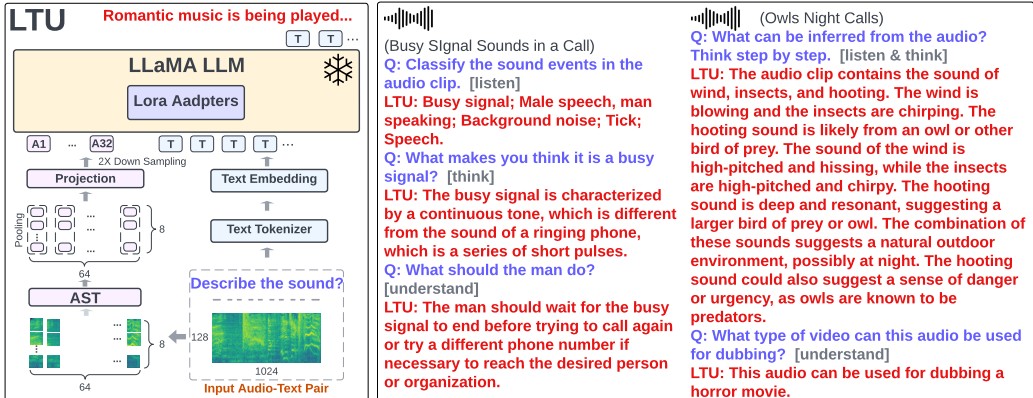

Figure 1: The `LTU` architecture and real samples showing how it listens, thinks, and understands.

The potential synergy between conventional audio models and LLMs in sound perception and reasoning motivates us to integrate them into a single model that is able to Listen to, Think about, and Understand the sound environment; we call this new model `LTU`. Specifically, the `LTU` framework effectively integrates a high-performant audio perception model AST (Gong et al., 2021b) with an open-source LLM LLaMA (Touvron et al., 2023). Since the data required to train `LTU` did not exist, we created a new dataset called `OpenAQA-5M` that combines 8 mainstream audio datasets, and formats all data as (audio, question, answer) tuples. Notably, `OpenAQA-5M` includes 3.7 million free-form open-ended and very diverse audio question-answer (AQA) pairs generated by a new GPT-assisted method called *Audio Instruction Generation (AIG)*.

Performance-wise, on audio classification tasks, `LTU` outperforms the conventional audio-text CLAP model (Elizalde et al., 2023) on all 8 benchmarks with an average relative improvement of 23.6%. In addition, unlike CLAP, `LTU` does not require a predefined label set during inference. On captioning tasks, `LTU` is comparable with SOTA specialized models. Of greater interest, `LTU` exhibits emerging audio reasoning and comprehension abilities (shown in Figure 1 and Table 6) that are absent in existing audio models. It can answer open-ended questions about audio with an instruction following and factual correctness rate of 82.9%. With these capabilities, `LTU` can serve as a foundation audio reasoning model. To the best of our knowledge, `LTU` is the first multimodal LLMs that focuses on general audio (rather than just speech) understanding, and our contributed `OpenAQA-5M` is the largest and most diverse audio question-answering dataset to date.

## 2 LTU MODEL ARCHITECTURE

The architecture of the `LTU` model is depicted in Figure 1.

**Audio Encoder.** We use an Audio Spectrogram Transformer (AST) (Gong et al., 2021b) pretrained with the CAV-MAE objective (Gong et al., 2022a) and finetuned on AudioSet-2M as our audio encoder. Each 10-second audio waveform is first converted to a sequence of 128-dimensional log Mel filterbank (fbank) features computed with a 25ms Hanning window every 10ms. This results in a 1024 (time) × 128 (frequency) spectrogram. We then split the spectrogram into 512 (64 (time) × 8 (frequency)) square patches of shape 16 × 16 and input them to AST. The output of AST is a sequence of 512 audio embeddings of 768-dimensions. We reshape it back to 64 (time) × 8 (frequency) and apply a frequency mean pooling and 2× temporal downsampling to produce a sequence of 32 audio embedding in temporal order (3.2Hz). Finally, we project the 768-dimensional audio embedding to 4096-dimensional to match the embedding dimension of LLaMA. The 32 audio embeddings are then concatenated with text embeddings as the input to the large language model.

**LLaMA Large Language Model.** In this paper, we use the LLaMA-7B large language model (Touvron et al., 2023) with Vicuna (Chiang et al., 2023) instruction following training. LLaMA is pretrained on a combination of natural and programming language corpora in a self-supervised manner. Based on LLaMA, Vicuna is further trained on instruction-following language prompts generated by GPT models (Taori et al., 2023; Peng et al., 2023a), and thus is more effective on a number of reasoning and generation tasks.

**Low-rank Adapters.** During `LTU` training, we adopt Low-rank Adaptation (Hu et al., 2021) (LoRA) rather than end-to-end fine-tuning of LLaMA parameters to 1) mitigate catastrophic forgetting (Goodfellow et al., 2013) and overfitting, and 2) improve the training efficiency. LoRA introduces a small set of auxiliary learnable weights, named LoRA adapters, on top of the pre-trained LLM, leaving all parameters in the latter unchanged. Each LoRA adapter is attached to a specific model layer, modifying its frozen parameter by the simple addition of a low-rank learnable matrix of the same size. In `LTU`, we inject LoRA adapters (rank=8 and $\alpha$=16) to the projection layers for key and query in all self-attention layers (Vaswani et al., 2017) of the LLaMA model, introducing only 4.2 million learnable parameters for fine-tuning.

**Training Objective.** LTU is trained on the next token prediction task conditioning on the past tokens and the reference audio, i.e., maximizing

$$P(x_t \mid x_{1:t-1}, A) \tag{1}$$

through cross-entropy for all $1 < t \leq T$ given text sequence $x_{1:T}$ and the reference audio $A$.

**Generation Setting.** Throughout this paper, we use a plain generation setting of Temperature=0.1, Top K=500, and Top P=0.95 with a repetition penalty of 1.1 (Fan et al., 2018; Keskar et al., 2019). We observe that this setting generally works well for all tasks; tuning generation parameters might benefit a specific task but is less practical in real applications.

## 3  THE OPENAQA DATASET

Table 1: The statistics of the `OpenAQA` dataset.

**Closed-Ended Audio Question Answering Data (1.9M)**

| Dataset | Task | | | | # Audio Samples | # QA Pairs | Percentage |
|---|---|---|---|---|---|---|---|
| | Classification | Acoustic Features | Caption | Temporal | | | |
| AudioSet-Strong | x | x | x | x | 102K | 683K | 12.0% |
| AudioSet | x | x | - | - | 500K | 538K | 9.5% |
| VGGSound | x | x | - | - | 184K | 367K | 6.5% |
| FSD50K | x | x | - | - | 41K | 82K | 1.4% |
| AudioCaps | x | - | x | - | 46K | 97K | 1.7% |
| Freesound | - | - | x | - | 91K | 91K | 1.6% |
| Clotho | - | - | x | - | 5K | 48K | 0.8% |
| Sound Bible | - | - | x | - | 1.2K | 12K | 0.2% |
| **Sum** | **345K (18.0%)** | **845K (44.1%)** | **430K (22.4%)** | **297K (15.5%)** | **845K** | **1,918K** | **33.8%** |

**Open-Ended Audio Question Answering Data (3.7M)**

| Dataset | Used Meta Information | | | | # Audio Samples | # QA Pairs | Percentage |
|---|---|---|---|---|---|---|---|
| | Audio Events | Acoustic Features | Caption | Temporal | | | |
| AudioSet-Strong | Multiple | x | Single | x | 91K | 901K | 15.9% |
| AudioSet-20K | Multiple | x | - | - | 19K | 184K | 3.2% |
| VGGSound | Single | x | - | - | 184K | 907K | 16.0% |
| FSD50K | Multiple | x | - | - | 41K | 403K | 7.1% |
| AudioCaps | Multiple | x | Multiple | - | 46K | 478K | 8.4% |
| Freesound | - | - | Single | - | 91K | 791K | 13.9% |
| Clotho | - | - | Multiple | - | 5K | 89K | 1.6% |
| Sound Bible | - | - | Single | - | 1.2K | 10K | 0.2% |
| **Sum** | | | | | **453K** | **3,764K** | **66.2%** |
| **Total** | | | | | **845K** | **5,682K** | **100.0%** |

Since LTU is a generative AI system and is anticipated to possess a wide range of capabilities for diverse audio tasks, existing datasets are not sufficient to train LTU. Therefore, we created a new audio dataset called `OpenAQA-5M`. Specifically, we formulate all tasks as audio question answering, i.e., each sample in the dataset is a `(audio, question, answer)` tuple, where `audio` and `question` are the model inputs, and `answer` is the label. This allows us to unify nearly all audio tasks into a single dataset and also maps the labels to a semantic space.

As there are already many high-quality audio datasets, we do not collect new audio data, but instead relabel existing public datasets including AudioSet (including a 500K subset of the original 2M weakly-labeled release (Gemmeke et al., 2017) and the 100K subset with temporally-strong labels (Hershey et al., 2021)), VGGSound (Chen et al., 2020a), FSD50K (Fonseca et al., 2021),

Table 2: Sample training audio question-answer pairs in `OpenAQA`.

| Closed-Ended Question-Answer Pairs |
|---|
| ***Audio Event Classification Based on Acoustic Features*** |
| Question: Classify the sound events in the audio clip based on acoustic features. |
| Answer: High-pitched, rapidly changing frequency → Ambulance (siren); Droning, loud and intrusive → Traffic noise, roadway noise; High pitched and loud → Accelerating, revving, vroom; Sharp and punchy → Generic impact sounds. |
| ***Audio Captioning*** |
| Question: Write an audio caption describing the sound. |
| Answer: An ambulance siren echoes while traffic noise fades, and an engine revs. |
| ***Audio Temporal Analysis*** |
| Question: Classify the sound events in the audio clip, also output the timestamp of each audio event. |
| Answer: Ambulance (siren): [0.0s-1.0s]; Accelerating, revving, vroom: [2.0s-10.0s]; Generic impact sounds: [6.7s-6.8s]... |
| **Open-Ended Question-Answer Pairs** |
| **GPT-3.5-Turbo Prompt:** Based on the following audio clip, generate 10 different types of complex open-ended questions that require step-by-step thinking, and corresponding answers. The following information is provided: the sound events appear in the audio clip, together with its acoustic features, and corresponding onset and offset time stamps. A description of the content of the audio clip is also provided. Questions should be about the audio, e.g., which sound event is recognized and why (e.g., based on its acoustic feature), what can be inferred based on the combination of sound events; the temporal relationship between the sound events and what can be inferred from that; the potential scenario that such an audio clip could happen, if the audio clip is special (e.g., urgent, funny, interesting, abnormal, unique, etc) and why, what mood or atmosphere this audio clip conveys, etc. The more complex and diverse the question, the better. (shortened, see Section M.) **Audio Meta-Information:** Sound Events: Sound of Ambulance (siren) (High-pitched and wailing): [0.0s-1.0s]; Sound of Traffic noise, roadway noise (Droning, loud and intrusive): [0.0s-10.0s]; Sound of Accelerating, revving, vroom (High-pitched, short and intense): [2.0s-10.0s]; Sound of Generic impact sounds (Loud and sharp): [6.7s-6.8s]. Description: An ambulance siren echoes while traffic noise fades, and an engine revs. |
| Question: How would you describe the tone of the sound of the accelerating engine? |
| Answer: The tone of the sound of the accelerating engine is high-pitched, short and intense. |
| Question: What is the acoustic feature that distinguishes the sound of the ambulance siren from the generic impact sounds? |
| Answer: The acoustic feature that distinguishes the sound of the ambulance siren from the sound of generic impact sounds is that the former is high-pitched and wailing, while the latter is loud and sharp. |
| Question: What can be inferred from the fact that traffic noise fades while the ambulance siren echoes? |
| Answer: It can be inferred that the ambulance is approaching or near the location where the audio clip was recorded, and that the traffic is yielding to the emergency vehicle. |
| Question: What mood or atmosphere does the audio clip convey? |
| Answer: The audio clip conveys a sense of urgency, due to the sound of the ambulance siren... |

AudioCaps (Kim et al., 2019), Freesound (Font et al., 2013), Clotho v2 (Lipping et al., 2019), and Sound Bible (soundbible.com, 2006) as our training data. For all datasets, we only include data marked as training and validation samples and exclude any data marked as test or evaluation. A total of 845K unique audio clips are used.

The `OpenAQA` dataset consists of two subsets: a 1.9M closed-ended question subset, and a 3.7M open-ended question subset, both are crucial to the success of LTU training. Although the LLaMA-7B large language model in LTU already has general knowledge and reasoning ability about the sound, in data generation, we use a larger and more powerful GPT-3.5-Turbo (Brown et al., 2020) LLM to improve the data quality. Table 1 presents the statistics of `OpenAQA-5M`.

### 3.1 CLOSE-ENDED AUDIO QUESTION-ANSWER GENERATION

We include four closed-ended audio tasks. For each task, we paraphrase the question with GPT-3.5-Turbo assistance to generate a diverse question set, but the answers are generated with a rule-based algorithm, and thus have a fixed format. The upper section of Table 2 presents samples of closed-ended question-answer pairs. As we will discuss in Section 4, such closed-ended tasks are crucial for guiding `LTU` when being conditioned on audio, and can greatly mitigate the hallucination issue.

**Classification**: The question for this task is "classify the sound events in the audio clip" and its GPT-assisted paraphrases; The answer is a list of sound class names that the audio clip contains.

**Acoustic Features**: This task aims to train the model to recognize low-level features for better generalization. Original audio datasets do not have this information. We thus generate the acoustic feature description using GPT-3.5-Turbo with the prompt "describe the acoustic characteristic of {sound class name} sound precisely with a sentence less than 10 words". We generate 10 different descriptions for each sound class. The question for this task is "classify the sound events in the audio clip based on acoustic features" and its GPT-assisted paraphrases. The answer is a list of GPT-3.5-Turbo-generated acoustic features and the sound class names.

**Captioning**: The question is "write an audio caption describing the sound", and the answer is the caption. For AudioCaps (Kim et al., 2019) and Clotho v2 (Drossos et al., 2020), we use the original manually-crafted captions. For Freesound (Font et al., 2013) and SoundBible (soundbible.com, 2006), we use the captions generated with GPT assistance in the WavCaps project (Mei et al., 2023).

**Temporal Analysis**: We use the time stamp information in the temporally strongly-labeled AudioSet (Hershey et al., 2021) to craft questions including listing the time stamps of all sound events, asking the time stamps of a specific sound, and the order of sounds with a rule-based algorithm.

## 3.2 OPEN-ENDED AUDIO QUESTION-ANSWER GENERATION

Compared to closed-ended QA, generating large-scale, diverse open-ended QA is more challenging. On the one hand, generating millions of QA pairs presents an impractical task for humans, even with crowd-sourcing; on the other hand, rule-based automatic generation methods could result in significant limitations in terms of output diversity (Abdelnour et al., 2022). For pure language tasks, it was found that using GPT large language models to automatically generate question-answer pairs as the training data for instruction tuning (Taori et al., 2023; Peng et al., 2023a) is a plausible solution.

Nonetheless, even the most advanced large language models (e.g., GPT-4) have not yet supported non-speech audio input (which is one of the main motivations of this work). How can we therefore use large language models to automatically generate audio QA? In this paper, we present *Audio Instruction Generation* (AIG). Specifically, we first extract the meta information of the audio (e.g., audio events, acoustic features, caption, and temporal information) and input the audio meta information to the GPT-3.5-Turbo model in the form of pure text, and then use the prompt shown in Table 2 to let the GPT-3.5-Turbo model generate audio QA pairs. In this process, GPT-3.5-Turbo only sees audio meta information instead of listening to the audio sample. We generate multiple QAs for each audio clip. With more meta-information provided, the generated QAs are more diverse, thus we generate more questions for samples with richer meta-information (e.g., strongly labeled AudioSet). We also use all 8 audio datasets to increase audio diversity.

A total of 3.7 million QA pairs are generated, which is about two orders of magnitude greater than the instruction tuning datasets used for pure language tasks (Taori et al., 2023). We observe consistent performance improvement with the increase of training data, indicating the necessity of having a larger dataset for audio since the LLaMA model has not been pretrained with any audio data. The generated open-ended QA pairs are very diverse, among the 3.7M QA pairs, 78% questions and 95% answers are unique, i.e., *most questions and answers only appear once in the dataset*. In addition, the generated QAs cover a wide spectrum from low-level tasks (e.g., identifying the audio events) to high-level understanding tasks (e.g., identifying the atmosphere of the audio). It is also worth mentioning that approximately 6.5% of the questions generated by GPT-3.5-Turbo do not have an answer based on the provided audio information, and the corresponding generated answers are "it cannot be determined from the audio that ...". These questions are equally valuable and important since they teach the model to know what it does not know and reduces the hallucination issue.

Note that only the raw audio and generated QA pairs are input to `LTU` in training, the audio meta information is only used in QA generation and is not input to `LTU`. Thus, the `LTU` model is forced to learn directly from the raw audio that contains richer and more fine-grained information rather than from the extracted meta-information. In inference, `LTU` *solely* use raw audio to answer the question.

## 4 LTU TRAINING RECIPE

We find an appropriate curriculum is crucial to successfully train `LTU`. As we mentioned in Section 2, we freeze the LLaMA model to mitigate catastrophic forgetting, and only train the AST audio encoder, the audio projection layer, and the LoRA adapters. Since the audio projection layer is randomly initialized, we first freeze AST and LoRA adapters and only train the audio projection layer with the closed-ended classification and acoustic feature description tasks in stage 1. Then in stages 2-4, we set all parameters (excluding LLaMA) as trainable, but gradually increase the openness of the

Table 3: The perception-to-understanding curriculum.

| Stage | Tr. Params | Tr. Task | # Tr. Samples | LR | # Epochs |
|---|---|---|---|---|---|
| 1 | Proj. | Clf. + Desc. | 1.2M | 1e-3 | 2 |
| 2 | All | Clf. + Desc. | 1.2M | 1e-4 | 2 |
| 3 | All | Closed-Ended | 1.9M | 1e-4 | 1 |
| 4 | All | All | 5.6M | 1e-4 | 1 |

training task, from only the classification task and acoustic feature description tasks (stage 2), to all closed-ended tasks (stage 3), and finally to all closed-ended and open-ended tasks (stage 4).

The reason for this design is that we find open-ended tasks to be too difficult for the model at the beginning of training. The model tends to use its language capability to answer the free-form open-ended question instead of conditioning on the audio input (i.e., hallucination). Thus, it is crucial to guide the model to attend to audio using closed-ended tasks with objective answers (e.g., sound class labels) in earlier stages, where the model gets a high penalty for wrong predictions. In other words, this curriculum teaches the model first to perceive, and then understand the sound. We refer to this training curriculum as a perception-to-understanding curriculum. As we will show in Section 5, the model performance significantly drops without such a training curriculum.

In all training stages, we use a batch size of 256 and linear learning rate decay with warmup. We set the text token cutoff length to 108. The model is trained on $4\times$ RTX A6000 GPUs for about 3 days.

## 5 EXPERIMENTS

### 5.1 CLOSED-ENDED AUDIO TASK EXPERIMENTS

Correct perception is the basis of advanced reasoning. If the model cannot correctly recognize the sounds, its answer to open-ended questions would be nothing but just hallucination. Therefore, before we discuss the exciting open-ended task results, we first rigorously evaluate LTU on classic closed-ended tasks including 8 audio classification benchmarks and 2 audio captioning benchmarks.

**Audio Classification:** Since LTU directly outputs audio label names or descriptions in text form instead of label index, to benchmark and compare it with existing models, we encode the LTU output and the evaluation dataset label names using a text encoder (gpt-text-embedding-ada), and compute the cosine similarity between the text embedding of LTU output and each label. For single-label classification tasks, we use the label that has the highest similarity score as the prediction and report accuracy or F1-score; for multi-label classification tasks, we use the similarity score as the prediction score and report the mAP. Note that LTU learns to only output prominent sound classes, which aligns with human perception. Conventional audio tagging models (e.g., AST) trained with binary cross entropy losses output a score for each class, to output prominent classes, a threshold needs to be calibrated for each class. However, this nice feature of LTU leads to a lower mAP score because the likelihood of non-prominent sound classes is not predicted. We tested two prompts "classify the sound events in the audio clip" and "write an audio caption describing the sound", while both led to good results in our subjective evaluation, the latter led to better text embedding for the automatic evaluation framework and is used for benchmarking. As shown in Table 4, LTU outperforms contrastive audio-text model CLAP (Elizalde et al., 2023) on all eight classification tasks with an average relative improvement of 23.6% and is comparable to a concurrent work (Wu et al., 2023b). Note that LTU is trained and evaluated with audios sampled at 16kHz while both CLAP and (Wu et al., 2023b) use 44.1KHz sampling rate, a higher sampling rate is known to lead to better audio classification performance but is also more computationally expensive. In addition, unlike CLAP and (Wu et al., 2023b), LTU does not need a pre-defined label set for classification.

**Audio Captioning:** We use the prompt "write an audio caption describing the sound" and take the model output as the prediction. We use SPICE (Anderson et al., 2016) as the evaluation metric and evaluate LTU on the test set of two major audio captioning benchmarks Clotho v2 (Drossos et al., 2020) and AudioCaps (Kim et al., 2019) with the standard evaluation protocol. As shown in Table 4, LTU achieves 17.0 and 11.9 SPICE scores on AudioCaps and Clotho, respectively, which is comparable with SOTA models. Note that compared with specialized models trained and evaluated on the same dataset, LTU is trained with more diverse data and thus may not fit the data-specific vocabulary or style, so the SPICE score may underestimate the performance of LTU as it does not count synonyms as correct, e.g., LTU's prediction of "a drill is running and vibrating" gets a 0 SPICE score though it is semantically close to the ground truth caption "a tool buzzing".

**Ablation Studies (Table 5):** First, we train LTU models with full finetuning (i.e., trainable LLaMA) and various LoRA adapter settings. When the learning rate is set to an appropriate value (2e-5), LTU performs slightly better on audio tasks but dramatically loses its original language reasoning ability, i.e., catastrophic forgetting, which justifies the need to freeze the LLaMA model. LoRA hyper-parameters do not have a large impact on model performance, but removing LoRA adapters will lead to noticeable lower performance, which justifies the need to have adapters for audio tasks.

Table 4: Closed-ended audio task performance. ZS: Zero-shot evaluation, the entire dataset is not seen in the training; ZS-: Weak zero-shot evaluation, the dataset is not used in training, but it is sourced from the same project as part of the training data. † Mean average precision (mAP) underestimates the performance of LTU as LTU does not predict the likelihood of non-prominent sound classes. ‡ Use a higher sampling rate of 44.1KHz than LTU (16KHz).

| Model | ESC50 (Acc) | DCASE (Mi-F1) | VS (Acc) | TUT (Acc) | BJO (Acc) | VGG (Acc) | FSD† (mAP) | AudioSet† (mAP) | Classif. Avg. | AudioCaps (SPICE) | Clotho (SPICE) | Cap. Avg. |
|---|---|---|---|---|---|---|---|---|---|---|---|---|
| *Best specialized models trained supervisedly on each dataset. Not generalizable to unseen label sets and tasks.* | | | | | | | | | | | | |
| Best Supervised & Specialized | 97.0 | 64.6 | 98.0 | 74.6 | 97.5 | 59.5 | 56.2 | 47.3 | 74.3 | 17.7 | 13.5 | 15.6 |
| *CLIP-like audio-text model. Generalizable to unseen labels, but a pre-defined label set is required for inference.* | | | | | | | | | | | | |
| AudioCLIP (Guzhov et al., 2022) | 69.4 | - | - | - | - | - | - | 25.9 | - | - | - | - |
| Wu et al. (2023b) ‡ | 89.1 | - | - | - | - | - | - | - | - | - | - | - |
| CLAP (Elizalde et al., 2023) ‡ | 82.6 | 30.0 | 49.5 | 29.6 | 47.5 | - | 30.2 | 5.8 | 40.7 | - | - | - |
| *Text generation model: One single model for all tasks. Directly output label names, no pre-defined label set is needed at inference.* | | | | | | | | | | | | |
| LTU | 83.1$^{ZS-}$ | 45.9$^{ZS-}$ | 55.6$^{ZS}$ | 32.5$^{ZS}$ | 69.9$^{ZS}$ | 50.3 | 46.3 | 18.7 | 50.3 | 17.0 | 11.9 | 14.5 |

Table 5: Ablation studies on LTU training settings (left) and training curriculum settings (right).

| Training Setting | Avg. Classif. Performance | Pure Language Task Instruction Following Rate |
|---|---|---|
| LoRA, r=8, LR=1e-4 (Default) | 50.3 | 87.6 |
| No LoRA Adapters | 38.3 (-12.0) | 88.1 (+0.5) |
| LoRA, LoRA_r=4 | 50.2 (-0.1) | 89.5 (+1.9) |
| LoRA, LoRA_r=16 | 50.5 (+0.2) | 87.7 (+0.1) |
| Full FT, LR=1e-4 | 50.4 (+0.1) | 35.0 (-52.6) |
| Full FT, LR=2e-5 | 51.3 (+1.0) | 53.5 (-34.1) |
| Freeze Audio Branch | 33.7 (-16.6) | 72.5 (-15.1) |

| Curriculum Setting | Avg. Classif. Performance |
|---|---|
| LTU (trained w/ default curriculum) | 50.3 |
| No Curriculum (same total tr. iterations) | 23.0 (-27.3) |
| No Open-ended Training (Stage 1&2&3 only) | 37.1 (-13.2) |
| No Open-ended Training (same total tr. iterations) | 47.3 (-3.0) |
| Use a large learning rate of 1e-3 in Stage 4 | 49.5 (-0.8) |
| Add half an epoch to the training in Stage 4 | 50.3 (-) |
| Train for only one epoch in Stages 1&2 | 49.7 (-0.6) |
| Cut tr. iterations by 50% across all stages | 45.3 (-5.0) |

Second, we train a model without the perception-to-understanding curriculum. To make a fair comparison, we train this model with the same number of total iterations as the LTU model. The audio task performance significantly declines, which justifies the necessity of the training curriculum. We also tested various training epochs and learning rates and found that the default setting is the best.

Third, we evaluate the performance of the model trained with only closed-ended tasks (i.e., stage 1-3) and find its performance on closed-ended tasks is worse than LTU trained with both closed-ended and open-ended tasks, indicating that open-ended task training also benefits closed-ended task performance, i.e., *learning to understand improves the perception ability*.

**Summary:** As an open-ended AQA model, LTU also exhibits a strong generalization ability across closed-ended datasets and tasks. Notably, LTU directly outputs its prediction in text, automatically filters out non-prominent sounds in the prediction, and requires no pre-defined label set for inference. These distinguish LTU from existing audio-text models and make it a potentially more attractive system for real-world applications.

## 5.2 OPEN-ENDED AUDIO TASKS EXPERIMENT

### 5.2.1 EMERGING AUDIO REASONING CAPABILITIES IN OPEN-ENDED TASKS

The main advantage of LTU is its capability to address open-ended audio tasks where LTU exhibits emerging reasoning capabilities that are not present in conventional models. In Table 6, we show LTU's answers to questions asking about realistic test audios. Key findings are as follows:

**LTU can answer follow-up questions about the details.** Conventional audio tagging models map audio to a set of pre-defined sound labels, but they cannot elaborate. In contrast, LTU can describe the details of the sound if the user asks a follow-up question. For example, in Table 6, Sample 1, LTU can tell us the bell rings *three times*, and has a feature of *bright and high-pitch*.

**LTU can explain its predictions.** Conventional models produce predictions without explanation. In contrast, LTU can explain its prediction if the user asks. For example, in Table 6, Sample 2, LTU tells the reason why it thinks the atmosphere is dangerous. More interestingly, in multi-turn conversation, it can explain why it made a wrong prediction when the user points out a mistake.

Table 6: LTU open-ended AQA demos (audios are available in the supplementary materials).

**Sample 1. Bell Ringing Sound**

Question: How many times the bell is rung?   [Follow-up Question, Detailed Perception Question]
Answer: The bell is rung three times.
Question: Describe the sound of the bell in this audio clip in detail.   [Follow-up Question, Detailed Perception Question]
Answer: The bell in this audio clip has a bright and ringing sound with a high pitch and short duration. It is a single ringing sound that is repeated several times in quick succession, creating a rhythmic pattern.
Question: What type is the bell?   [LTU knows what it does not know.]
Answer: The audio clip does not provide enough information to determine the type of the bell. It could be a church bell, a school bell, or any other type of bell that produces a similar sound.

**Sample 2. Gun Shot Sound**

Question: Describe the audio clip with a sentence.   [Closed-Ended Question]
Answer: Gunshots and footsteps are heard, followed by a man speaking and a beep.
Question: What is the possible scenario of this sound?   [Scene Understanding]
Answer: It is possible that the man is walking through a war zone or a dangerous area.
Question: What mood or atmosphere does this audio clip convey? Explain Why.   [Audio Mood Understanding]
Answer: This audio clip conveys a tense and potentially dangerous atmosphere, as the gunshots and the man's speech suggest that there is a potential threat or conflict.
Question: If I hear the sound, what should I do?   [Action Suggestion]
Answer: If you hear the sound, you should immediately seek shelter and call for help...

**Sample 3. Owls Night Calls**

Question: What sound is heard?   [Closed-Ended Question]
Answer: The sound of wind is heard throughout the entire clip, while the sound of insects, hoot, and owl are heard intermittently and at different time stamps.
Question: What insect is heard? Why?   [Prediction Explanation]
Answer: The insect heard is a cricket. It is high-pitched and chirpy, which is why it is recognized.
Question: Is this audio more likely to be recorded in the morning or evening? Why?   [Scene Understanding]
Answer: It is more likely to be recorded in the evening because the owl hooting sound is more prominent during the night.
Question: What type of video can this audio be used for dubbing?   [Audio Usage Suggestion]
Answer: This audio can be used for dubbing a horror movie.

**LTU can think step by step.** Conventional models produce predictions without intermediate steps. In contrast, LTU can do step-by-step reasoning if the user asks. For example, in Figure 1, in the Right Sample, LTU does advanced step-by-step reasoning to understand the audio clip.

**LTU can understand the scene and can connect sounds to actions and usages.** In Table 6, Sample 2, LTU understands it is a dangerous scene and suggests seeking shelter when a gunshot is heard; in Sample 3, LTU suggests the owl calling sounds can be used for dubbing a horror movie.

**LTU knows what it does not know.** Though we would like LTU to answer all our questions, we do not want it to hallucinate what it does not hear clearly or answer questions that are not answerable based on the audio. As shown in Table 6, Sample 1, LTU refuses to give a definite answer about the bell type but instead lists a few possibilities. We find the ∼6.5% unanswerable QA pairs in the training data improve LTU's ability to refuse unanswerable questions. If we remove them from the training set, LTU has a lower refusal rate to GPT-generated unanswerable questions (69.3%→51.9%), i.e., a higher risk of hallucination. Also, the close-ended performance slightly drops (50.3→50.0). Note that GPT-generated unanswerable questions are not answerable by GPT based on the audio meta information, but they may be answerable by LTU based on the audio file.

### 5.2.2 QUANTITATIVE EVALUATION

In addition to the compelling qualitative analysis presented in Section 5.2.1, we further conducted a series of evaluations to quantitatively assess the overall quality of LTU predictions.

We evaluated the ability of LTU to directly answer any question raised by the user (i.e., instruction following) and if the answer is factually correct rather than a hallucination. Owing to the absence of an automated evaluation protocol for open-ended audio question answering, we conducted a two-stage human subjective evaluation through Amazon Mechanical Turk (AMT) with a total of 476 unique human evaluators independent of us. In Stage 1, we use open-ended questions generated by GPT-4 to evaluate LTU (note that in generating our training data we only used GPT 3.5). We play the audio and show the human evaluators the question and the LTU answer and ask them to respond to each of the following questions: 1) if the answer directly addresses the question (instruction following); 2) if LTU answer is factually correct; 3) compare between LTU and GPT-4 answers, which is better? Finally, we ask the human evaluators to ask another question and provide an answer

based on the audio, which are used in Stage 2 evaluation. Stage 2 is the same as Stage 1 except we use human-generated QA pairs, i.e., `LTU` answers questions posed by humans and not by GPT-4, and other human evaluators compare its predictions with human-generated responses.

Table 7 presents the results of the evaluation. The instruction following & factual correct rate of LTU is 82.9% to human-generated audio questions, and 75.9% to GPT-4 generated audio questions, respectively. In addition, 74.9% of human evaluators rate LTU answers are better than human-generated answers, and 69.1% of human evaluators rate LTU answers are better than GPT-4 generated answers. As a supplement to the human subjective evaluation, we follow (Peng et al., 2023a) to automatically evaluate the instruction fol-

Table 7: Human evaluation settings and results.

|  | Stage 1 | Stage 2 |
|---|---|---|
| #Questionares | 615 | 737 |
| #Unique Human Evaluators | 202 | 362 |
| Evaluation Audio Source | AudioSet Evaluation Set | |
| Evaluation QA Generated by | GPT-4 Based on Audio Meta Info | Humans Based on Listening to Audios |
| LTU Instruction Following Rate | 87.1% | 90.0% |
| LTU Factual Correctness Rate | 79.3% Correct 14.6% Partially Correct | 83.7% Correct 8.4% Partially Correct |
| LTU Instruction Following & Factual Correctness Rate | 75.9% | 82.9% |
| Compare LTU Output with GPT-4 or Human Answer | 69.1% Prefer LTU over GPT-4 | 74.9% Prefer LTU over Human Answer |

lowing rate with GPT-4 assistance with the prompt "Below is a pair of question and response. Identify if the response answers the question". Results show that `LTU` has an instruction following rate of 96.9%. In contrast, the model trained with only closed-ended tasks cannot follow instructions well (22.5%). These results suggest `LTU` indeed does consistently well for open-ended tasks.

## 6    RELATED WORK

In the past year, modern large language models (OpenAI, 2023; Chiang et al., 2023; Zhang et al., 2023b) have demonstrated powerful reasoning and understanding abilities. To further enable multi-modal capabilities for LLMs, Flamingo (Alayrac et al., 2022), GPT-4 (OpenAI, 2023), Kosmos (Huang et al., 2023b), BLIP (Li et al., 2023), LLaMA-Adapter (Zhang et al., 2023b), PaLM-E (Driess et al., 2023), and LLaVA (Liu et al., 2023) each proposes a different integration method. Nonetheless, they all focus on the visual modality. Among these, LLaVA (Liu et al., 2023) (concurrent) is the closest work with us as it also proposes an instruction tuning method and a 2-stage training. However, the main motivation and implementations are different due to the very different focusing modalities. In the sound domain, WavPrompt (Gao et al., 2022) introduces a framework that integrates GPT-2 and a speech representation model to address speech understanding in the classification form. SpeechPrompt (Chang et al., 2022; 2023) studies speech understanding in a text-less style through prompt tuning spoken language model (Lakhotia et al., 2021). More recently, SpeechGPT (Zhang et al., 2023a) (concurrent) and Speech-LLaMA (Wu et al., 2023a) (concurrent) apply LLMs to speech recognition and translation, respectively. However, all these efforts mainly focus on speech and have limited ability for open-ended question answering. AudioGPT (Huang et al., 2023a) (concurrent) introduced a system using Chat-GPT as an interface for a broad collection of audio and speech applications and thus relies on external audio systems. Pengi (Deshmukh et al., 2023) (concurrent) also proposes an audio language model, but is less focused on open-ended tasks. `LTU` differs from these prior works in that 1) `LTU` focuses on general audio understanding instead of pure speech; 2) `LTU` can answer both closed-ended and open-ended questions, and 3) `LTU` is a standalone model that does not rely on external systems. To the best of our knowledge, `LTU` is the first model bridging audio perception and advanced reasoning.

## 7    CONCLUSION

This paper aims to build an artificial intelligence system named `LTU` that can listen to, think about, and understand the surrounding audio environment. With extensive experiments, we find a dataset consisting of both closed-ended and open-ended questions and a perception-to-understanding training curriculum are two key components to successfully train `LTU`. The closed-ended tasks are used to train the model at the beginning stage to force the model conditioned on audio, while the open-ended tasks enable the model to have advanced reasoning and understanding ability. By formulating all tasks as audio question-answering, we can train `LTU` with many diverse tasks. Consequently, `LTU` not only outperforms the SOTA audio-text model CLAP with an average relative improvement of 23.6% on closed-ended tasks, but also exhibits emerging reasoning capability in open-ended tasks.

**Limitations:** In this work, we mainly focus on general audio understanding and `LTU` thus has limited ability to understand speech content, i.e., it is not an automatic speech recognition model.

ACKNOWLEDGMENTS

This research is supported by the MIT-IBM Watson AI Lab. We thank Samuel Thomas, Hilde Kuehne, Rogerio Feris, and Brian Kingsbury for their helpful discussions.

ETHICS STATEMENT

The audio data used in this paper are publicly available online, we do not use private audio data to train the model. The proposed audio large language model has the potential to benefit individuals with disabilities by providing hearing aids. However, it is imperative to acknowledge the potential applications of such models in security-related domains. While these models inherently do not specialize in speech recognition, their capabilities can, nonetheless, be repurposed or adapted for tasks that might have security implications. That said, given their primary design not being for speech recognition, the associated risk is comparatively mitigated. It is important for researchers, developers, and users who employ this technology responsibly to ensure its application aligns with ethical considerations and avoids potential misuse.

REPRODUCIBILITY STATEMENT

We document all implementation details in Section 2, 3, 4 and Appendix. Code, dataset, and pretrained models are available at https://github.com/yuangongnd/ltu.

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

## A  EXTENDED LITERATURE REVIEW

Due to the space limitation, in the main manuscript, we focus on discussing works closest to `LTU` (mainly LLM-related research). In this section, we further extend the discussion and focus more on general audio processing research.

In the past decade, **audio tagging and classification** research has moved from small and constrained datasets consisting of tens of sound classes to much larger datasets with a greater variety and range of real-world audio events. A significant milestone is the release of the AudioSet corpus (Gemmeke et al., 2017) containing over 2 million audio clips labeled with a set of 527 event labels. Based on AudioSet, a series of efforts have been made to improve the audio tagging performance in terms of mAP (Gong et al., 2021a;b; Ford et al., 2019; Wang et al., 2019; Yu et al., 2018; Kong et al., 2018; 2019; Kumar et al., 2018; Wu & Lee, 2018; Huang et al., 2022). Nonetheless, these models can only predict sound class in the 527-class AudioSet label set, and cannot generalize to unseen sound classes. To overcome this limitation, **zero-shot audio classification** has been proposed in multiple recent work (Xie & Virtanen, 2019; 2021; Xie et al., 2021; Gu et al., 2022; Lee et al., 2021; Elizalde et al., 2023; Guzhov et al., 2022). Specifically, zero-shot learning uses auxiliary information about the sound classes such as their textual descriptions and maps the audio to the auxiliary information instead of label indexes. In the inference stage, the model can generalize to new classes as long as its auxiliary information is provided. However, a predefined label set is still needed. `LTU` differs with the above efforts in that 1) it is not constrained with a fixed label set and can generalize to unseen classes; and 2) it directly outputs the label names in text form and does not require a predefined label set in the inference stage. These make `LTU` a more practical system for real-world applications.

On the other hand, a sound label (usually a word or a phrase) is not informative enough for many applications. To better describe the sound, **automated audio captioning** (AAC), the task to generate audio content description using free text, is getting increasingly more attention. Large-scale audio captioning datasets including AudioCaps (Kim et al., 2019) and Clotho (Drossos et al., 2020) established a solid basis for AAC research and many new approaches have been proposed including (Xu et al., 2020; Xie et al., 2022; Xu et al., 2021; Chen et al., 2020b; Koizumi et al., 2020; Mei et al., 2021; 2022). The main limitation of audio captioning is that the description is usually short (less than 20 words) and cannot cover all information of the audio. Therefore, the caption may miss the information that the user is interested in. To overcome this issue, a new task **audio question answering** has been proposed (Abdelnour et al., 2018; 2022). While it is an important step towards audio understanding and reasoning, both (Abdelnour et al., 2018; 2022) have a limited number of closed-ended question types and support a limited number of sound events. `LTU` differs from the above efforts in that `LTU` can not only generate audio caption and description, but also answer any *open-ended* questions about the details and explanation. With the LLaMA large language model, `LTU` exhibits advanced reasoning and understanding ability, e.g., it can connect the audio with actions. Such reasoning and understanding ability are not yet present in existing audio models.

## B  EXTENDED DISCUSSION OF LIMITATIONS

First, the LLaMA-7B model used in `LTU` is the smallest model in the LLaMA model family, LLaMA-13B, and larger models are more commonly used and may have a stronger reasoning and understanding ability. Particularly, larger LLM may exhibit emerging properties that the smaller LLM model does not have. While we show `LTU` exhibits stronger audio reasoning and understanding abilities than existing models, it could have been even stronger if a larger LLM was used. The choice of using LLaMA-7B is mainly due to the limit of our computational resources.

Second, for the same computational efficiency consideration, we applied a $2\times$ temporal downsampling to make the audio token frequency at 3.2Hz. This may limit the fine-grained temporal reasoning ability of the model. Increasing the temporal resolution of the audio input may further improve `LTU`.

Third, while we used 8 audio datasets to construct the `OpenAQA-5M` dataset, only a small portion of them are with temporal-level annotation (i.e., the onset and offset of the sound event), which is important information for GPT-3.5-Turbo to generate time-related questions. Therefore, the capability of the model to temporally localize the sound event might be weaker than its capability to recognize the sound events.

Finally, while we find closed-ended training and the unanswerable training questions greatly mitigate the hallucination issue (the model will answer "I do not know" to unanswerable questions), hallucination and bias may not be completely eliminated and may occur in some cases.

## C  EXTENDED DISCUSSION ABOUT THE CHOICE OF AUDIO ENCODER

In this work, we use an Audio Spectrogram Transformer (AST) pretrained (with the CAV-MAE objective) and finetuned on AudioSet-2M proposed in (Gong et al., 2022a) as our audio encoder. CAV-MAE is a self-supervised pretraining framework that combines contrastive learning and masked data modeling. In the CAV-MAE pretraining stage, AST and a visual branch are pretrained jointly with both audio and visual data of AudioSet, but in the finetuning stage, only the audio branch (i.e., AST) is finetuned and only audio data of AudioSet is used. Therefore, our AST audio encoder only takes audio as input. The main reasons why we use AST in CAV-MAE instead of vanilla AST (Gong et al., 2021b) are performance and efficiency. Performance-wise, the AST in CAV-MAE leads to an mAP of 46.6 on the AudioSet evaluation set, which is better than vanilla AST (45.9 mAP); Efficiency-wise, the AST in CAV-MAE splits the audio spectrogram into patches without overlapping and leads to a patch sequence length of 512 for each 10-second audio clip, which is much shorter than that of vanilla AST (1212). Due to the quadratic complexity of the Transformer, the AST in CAV-MAE is noticeably more efficient. In addition, due to the multi-modal CAV-MAE pretraining, we expect the representation of AST in CAV-MAE to be more semantic than vanilla AST. Note that while AST might be an optimal choice, any audio model can serve as the audio encoder of `LTU` as long as it can map audio input to a continuous embedding space and can be trained with backpropagation.

## D  EXTENDED DISCUSSION ON THE IMPACT OF UNANSWERABLE QUESTIONS

Table 8: Impact of "unanswerable" questions in the training set.

| | Avg. Classif. Performance | Avg. Caption. Performance | Refusal Rate to Test Unanswerable Questions |
|---|---|---|---|
| Original LTU | **50.3** | **14.4** | **69.3%** |
| LTU Trained without Unanswerable Questions | 50.0 | 14.3 | 51.9% |

GPT naturally generates about 6.5% questions that do not have an answer based on the given audio meta information with the proposed AIG method. By default, we include them in the training set. As shown in Table 8, if we remove them from the training set, we find the close-ended performance would slightly drop, more importantly, it has a lower refusal rate to GPT-generated unanswerable questions, i.e., a higher risk of hallucination. Note that questions not answerable by GPT based on the audio meta information may be answerable by LTU based on the audio file, so this test is an estimation. This also shows an interesting phenomenon that even when there are almost no unanswerable QAs in the training data, LTU is still able to refuse to answer questions that it is not sure about, but with a lower refusal rate, i.e., this behavior seems to be partly learned in the LLM training stage.

# E  SCALING UP THE LARGE LANGUAGE MODEL

Table 9: Comparing the performance of `LTU` based on 7B LLaMA and 13B LLaMA

| LLaMA Size | Audio Classification | | | | | | | | | Audio Captioning | | | Instruction Following Rate for Open-Ended Questions | |
|---|---|---|---|---|---|---|---|---|---|---|---|---|---|---|
| | ESC50 | DCASE | VS | TUT | BJO | VGG | FSD | AudioSet | Classif. Avg. | AudioCaps | Clotho | Cap. Avg. | Audio Task | Language Task |
| 7B | 83.1 | **45.9** | 55.6 | **32.5** | **69.9** | 50.3 | 46.3 | **18.7** | **50.3** | **17.0** | 11.9 | 14.5 | **96.9** | 87.6 |
| 13B | **83.6** | 44.7 | **64.2** | 31.0 | 52.1 | **50.7** | **46.8** | 17.9 | 48.9 | 16.9 | **12.1** | **14.5** | 95.4 | **89.6** |

Throughout the paper, we use the LLaMA model consisting of 7 billion parameters (LLaMA-7B) as the LTU language model. In this section, we conduct experiments on `LTU` based on LLaMA with 13 billion parameters (LLaMA-13B). As shown in Table 9, we find the overall performance on audio classification slightly drops with the increased size of LLaMA while the captioning performance is slightly stronger. There is no obvious difference in the instruction following rate for pure language and audio open-ended questions. However, we find that the 13B model tends to output more rational answers due to its better knowledge base, as demonstrated in the following sample.

> **Question:** "Question: How would the audio clip be different if the power tool was being used in a larger or open space?"
>
> **LTU-7B:** "LTU-7B: The sound of the power tool would likely be louder and more noticeable in an open space." (incorrect)
>
> **LTU-13B:** "LTU-13B: If the power tool was being used in a larger or open space, the sound of the power tool would likely be more diffuse and less intense than it is in this clip." (correct)

# F  DISCUSSION ON THE AUDIO AND TEXT EMBEDDING SPACE ALIGNMENT

Table 10: Compare `LTU` models trained with various alignment modules and additional audio-text alignment training strategies.

| Model | Audio Classification | | | | | | | | | Audio Captioning | | | Instruction Following Rate for Open-Ended Questions | |
|---|---|---|---|---|---|---|---|---|---|---|---|---|---|---|
| | ESC50 | DCASE | VS | TUT | BJO | VGG | FSD | AudioSet | Classif. Avg. | AudioCaps | Clotho | Cap. Avg. | Audio Task | Language Task |
| Linear Layer w/ Alignment Training | **83.2** | 45.3 | **59.5** | 30.4 | 64.4 | 50.0 | **46.5** | 18.3 | 49.7 | 16.8 | **11.9** | 14.4 | 86.5 | 93.2 |
| Transformer Layer w/ Alignment Training | 80.6 | **46.8** | 55.8 | 30.5 | 61.9 | **51.9** | 45.7 | 16.7 | 48.7 | 16.8 | 11.5 | 14.2 | **89.9** | 93.1 |
| Linear Layer wo/ Alignment Training (Default) | 83.1 | 45.9 | 55.6 | **32.5** | **69.9** | 50.3 | 46.3 | **18.7** | **50.3** | **17.0** | 11.9 | **14.5** | 87.6 | **96.9** |

Throughout the paper, we utilize the AST with CAV-MAE pretraining and fine-tuning as our audio encoder. Prior to LTU training, this audio encoder has not been trained for any audio-text alignment tasks; that is, its output embedding is *not* in the same space as text embeddings. The AST's output is used as *soft prompts* for the large language model. According to Lester et al. (2021); Driess et al. (2023), these (multimodal) *soft prompts* do not necessarily need to be in the text embedding space. However, it remains unclear whether audio-text alignment pretraining prior to LTU training could enhance performance. Therefore, in this section, we conduct further experiments to investigate the impact of audio-text alignment pretraining.

Specifically, we add an additional audio-text alignment training stage before LTU training. we test two types of alignment modules: one is the linear layer used in the original LTU, and the other is a Transformer layer, which allows for more model capacity in the alignment. We train the alignment module on 1.6 million audio-text pairs (comprising audio labels or captions) for 10 epochs, using a batch size of 1536 distributed over 4 GPUs, with 384 samples on each GPU. We start with an initial learning rate of 1e-3 and halve it after each epoch. During this stage, we keep the audio and LLaMA text encoders frozen and only train the alignment module. We use the following loss:

$$\mathcal{L} = \mathcal{L}_{\mathrm{c}} + \lambda \cdot \mathcal{L}_{\mathrm{mse}}. \tag{2}$$

$$\mathcal{L}_{\mathrm{mse}} = \frac{1}{N} \sum_{i=1}^{N} (e_i^a - e_i^t)^2 \tag{3}$$

$$\mathcal{L}_{\mathrm{c}} = -\frac{1}{N} \sum_{i=1}^{N} \log \left[ \frac{\exp(s_{i,i}/\tau)}{\sum_{k \neq i} \exp(s_{i,k}/\tau) + \exp(s_{i,i}/\tau)} \right] \tag{4}$$

where $\lambda = 10$ is used to balance the scale between MSE loss and contrastive loss, $e^t$ and $e^a$ are the text and audio embeddings, respectively, $N = 384$ is the batch size, $s_{i,j}$ is the similarity score calculated as the dot product of normalized text embedding $e_i^t$ and audio embedding $e_j^a$, and $\tau = 0.05$ is the temperature. This loss aligns both the scale and direction of the audio and text embeddings.

We present the results in Table 10 and observe that for the same linear projection layer, additional audio-text alignment training does not significantly impact model performance. Furthermore, scaling up the alignment module to a Transformer layer even worsens the performance. We hypothesize that this is attributable to several reasons:

1. In the original `LTU` training, the model is trained with 5.6 million samples over multiple epochs, and the audio encoder is set as trainable for three stages of the training. Consequently, during the original LTU training process, the audio encoder is sufficiently trained to bridge the gap with the LLM, and additional alignment training is unnecessary.

2. Increasing the capacity of the alignment module would introduce a larger number of randomly initialized parameters, complicating the training process. Considering that the AST audio encoder is a 12-layer Transformer and was set as trainable in the original LTU training, an increase in the capacity of the alignment module is not necessary.

3. The audio embeddings serve as soft prompts (Lester et al., 2021; Driess et al., 2023) for LLMs and do not necessarily need to reside in the text embedding space. Therefore, additional audio-text training does not provide a better initialization for LTU training.

## G  COMPARE WITH PENGI

In this section, we compare `LTU` with Pengi (Deshmukh et al., 2023), a concurrent work on audio language models.

### G.1  TECHNICAL SIMILARITY AND DIFFERENCE

**Similarity**

1. Architecture-wise, both Pengi and `LTU` connect an audio encoder with an autoregressive language model. Both models can generate text from the given audio and text prompt.

2. Performance-wise, both Pengi and `LTU` can do zero-shot predictions on unseen datasets. On closed-ended tasks, Pengi and `LTU` perform similarly.

**Difference**

1. *Motivation*: Pengi focuses on *transfer learning*, i.e., using a single model for 8 audio perception tasks, while `LTU` focuses on *unifying audio perception with understanding*. Unlike `LTU`, Pengi is not designed for audio understanding and answering *free-form open-ended* questions from users.

2. *Language Model Scale*: Pengi employs GPT2-Base (124M parameters) as its language model, whereas `LTU` uses LLaMA (7B parameters), which is over 50 times larger than GPT2-Base. Additionally, LLaMA is trained with significantly more data than GPT2. Scaling up the language model substantially enhances `LTU`'s understanding capabilities.

3. *Training Data*: Pengi is trained on multiple audio datasets, and directly uses the original text-form labels as answers for AQA. This approach is similar to the closed-ended part of our `OpenAQA` dataset. However, `OpenAQA` also includes another 3.7M open-ended AQAs generated with the proposed *audio instruction generation* method (not generated from a template). The open-ended portion of our training data is crucial in enabling `LTU` to answer *any free-form open-ended* questions.

4. *Training Curriculum*: To better unifying audio perception and generation, `LTU` adopts a unique perception-to-understanding training curriculum.

5. *Open-Ended Evaluation*: `LTU` undergoes evaluation for both closed-ended and open-ended tasks, incorporating both subjective and objective evaluations. Conversely, Pengi is evaluated solely with closed-ended tasks. Please note that the definition of the open-ended task differs between the Pengi paper and this paper; Pengi considers audio captioning as an open-ended task, while we categorize them as closed-ended tasks.

## G.2 Closed-Ended Task Performance Comparison

Table 11: Pengi and `LTU` performance comparison on closed-ended audio classification tasks.

|  | ESC50 | FSD50K | DCASE | TUT | Beijing Opera | Vocal Sound | Average |
|---|---|---|---|---|---|---|---|
| Pengi | **91.9** | **46.7** | 33.8 | **35.2** | 62.3 | 60.3 | 55.1 |
| LTU (Default) | 83.1 | 46.3 | 45.9 | 32.5 | **69.9** | 55.6 | 55.6 |
| LTU (Full FT) | 85.9 | 45.7 | **47.2** | 33.0 | 59.8 | **69.0** | **56.7** |

We summarize the performance of `LTU` and Pengi on closed-ended audio classification tasks in Table 11. Pengi and `LTU` each win on three out of the six tasks. In general, `LTU` with the default LoRA setting performs similarly to Pengi (average score 55.6 vs 55.1), while the fully finetuned version of `LTU` performs slightly better, achieving an average score of 56.7.

## G.3 Open-Ended Task Performance Comparison

Table 12: Comparing the instruction following rate of Pengi and `LTU` on open-ended questions genearated by GPT-3.5-Turbo, GPT-4, and human. The instruction following rate is evaluated by GPT-4 with the prompt "Below is a pair of question and response. Identify if the response answers the question. Return yes or no."

|  | Open-Ended Questions | | |
|---|---|---|---|
|  | GPT-3.5 Generated Questions | GPT-4 Generated Questions | Human Generated Questions |
| Pengi | 7.4 | 3.1 | 28.3 |
| LTU | **95.9** | **96.9** | **70.1** |

We use GPT-4 (with the prompt "Below is a pair of question and response. Identify if the response answers the question. Return yes or no.") to evaluate the instruction following rate of `LTU` and Pengi in response to open-ended questions and present the results in Table 12. With a stronger language model and training with the OpenAQA dataset, `LTU` dramatically outperforms Pengi. However, it is important to note that Pengi is neither designed nor trained for open-ended question answering and audio understanding, making this task not fair to Pengi. We conduct this experiment just to highlight the differences between `LTU` and Pengi.

## G.4 Summary

To summarize, while `LTU` and Pengi share some similarities in architecture, they have different motivations and designs. They perform similarly in closed-ended audio classification tasks, but `LTU` is more capable of answering open-ended questions and demonstrating audio understanding.

# H LTU Temporal Analysis Experiments

In Section 5.1, we show the `LTU` model performance on closed-ended tasks of classification and captioning. In this section, we further evaluate `LTU`'s capacity in temporal analysis. Specifically, we conduct experiments on the audio event order recognition task (seen in the closed-ended training stage) and audio event counting task (unseen in the closed-ended training stage) with ESC-50 (Piczak, 2015), a dataset not being used in training. For both experiments, we only include the cases in which `LTU` follows the instruction.

## H.1 Recognizing the Order of Sound Events

We create a test set where each test sample consists of two audio samples from different categories of the ESC50 dataset, with no temporal overlapping. We then ask `LTU` to predict the order of the sound events with the prompt "Which sound begins and ends first?". We use regular expressions and cosine similarity (based on gpt-text-embedding-ada) to interpret the raw `LTU` output. For example, for the `LTU` raw output "the applause starts first, while the footsteps follow in a rhythmic pattern afterward.", we first extract "applause" and "footsteps follow in a rhythmic pattern" using regular expressions. Then we compare the cosine similarity between the text embedding of the ground truth "clapping" with the text embeddings of "applause" and "footsteps follow in a rhythmic pattern", respectively. Since the cosine similarity between the text embedding of the ground truth "clapping" and the `LTU` prediction "applause" is higher, we count this sample as correct.

We evaluate `LTU` on 550 such samples, obtaining 390 correct and 160 incorrect answers, achieving an accuracy of 70.9%. This shows that `LTU` can recognize the order of sound events with a reasonable accuracy.

Table 13: `LTU` sound event counting experiment results. We create a test set where each test sample consists of one or two (repeated) audio samples from the ESC50 dataset, with no temporal overlapping. We then ask `LTU` to count the sounds in the audio clips using the prompt, "In how many instances is the sound of {:s} heard?". Since one ESC50 audio sample may contain sounds multiple times (e.g., three dog barking sounds), we calculate the Pearson Correlation Coefficient (PCC) between the `LTU` output count and the number of ESC50 audio clips in the test sample, finding a high correlation for many sound classes. Additionally, we calculate the ratio between `LTU` output counts for test samples consisting of two ESC50 samples and test samples consisting of one ESC50 sample. Ideally, this value should be 2, and indeed, it is close to 2 for many sound classes. However, there are some sound classes that `LTU` cannot count well, potentially due to an undefined number of sound appearances for these classes. This demonstrates the sound event counting capability of `LTU`, although it has not been explicitly trained for this task in the closed-ended training stage.

| Sound Class | Pearson Correlation Coefficient (PCC) ↑ | LTU Output Count Ratio for Double vs. Single ESC50 Audio Samples (Ideal = 2) |
|---|---|---|
| Chainsaw | 0.81 | 2.5 |
| Brushing Teeth | 0.79 | 2.6 |
| Helicopter | 0.77 | 2.1 |
| Siren | 0.58 | 2.0 |
| ... | | |
| Train | 0.02 | 1.01 |
| Toilet Flush | -0.03 | 0.98 |
| Frog | -0.18 | 0.92 |

## H.2 Counting the Number of Appearances of Sound Events

In Table 6, the first sample shows a demo where `LTU` recognizes the bell rings *three times*. In this section, we further quantify `LTU`'s capacity in counting the number of appearances of sound events. Note that this task is not in our closed-ended training. One problem in evaluating this task is that the definition of the number of appearances is not clear for all sound classes, e.g., it is hard to define what the appearance of a train sound is as it could consist of multiple click-clack sounds. Thus, we create a test set where each test sample consists of one or two (repeated) audio samples from the

ESC50 dataset, with no temporal overlapping. We then ask `LTU` to count the sounds in the audio clips using the prompt, "In how many instances is the sound of {:s} heard?". Since one ESC50 audio sample may contain sounds multiple times (e.g., three dog barking sounds), we calculate the Pearson Correlation Coefficient (PCC) between the `LTU` output count and the number of ESC50 audio clips in the test sample. As shown in Table 13, we find a high correlation for many sound classes. Additionally, we calculate the ratio between `LTU` output counts for test samples consisting of two ESC50 samples and test samples consisting of one ESC50 sample. Ideally, this value should be 2 because the number of appearances of sounds in two ESC50 samples should be twice the number of appearances of sounds in one ESC50 sample. We find it is indeed close to 2 for many sound classes. However, there are some sound classes that `LTU` cannot count well, potentially due to an undefined number of sound appearances for these classes. This demonstrates the sound event counting capability of `LTU`, although it has not been explicitly trained for this task in the closed-ended training stage.

### H.3 SUMMARY

To summarize, `LTU` demonstrates reasonable ability in temporal analysis. However, compared with classification and captioning tasks, its temporal analysis ability is weak. We hypothesize that this is due to two reasons: First, we aggressively pool the audio tokens to save computation, which may hurt fine-grained tasks; Second, `LTU` has not been trained with a sufficient amount of data for temporal analysis, as reported in Peng et al. (2023b), such training is crucial for fine-grained tasks.

## I OPEN-ENDED EVALUATION DETAILS

### I.1 HUMAN SUBJECTIVE EVALUATION

As we mentioned in the main manuscript, we conducted a two-stage human evaluation through Amazon Mechanical Turk (AMT) with a total of 476 unique human evaluators independent of us. In Stage 1, we evaluate LTU with GPT-4 generated open-ended questions, we play the audio and show the human evaluators the question and LTU answer and ask them to rate 1) if the answer directly addresses the question (instruction following); 2) if LTU answer is factually correct; 3) between LTU and GPT-4 answers, which is better? Finally, we ask the human evaluators to ask another question and provide the answer based on the audio, which are used in Stage 2. Stage 2 is the same as Stage 1 except we use human-generated QAs. The actual questionnaire is shown in Figure 2.

To better control the quality of human subjective evaluation, we 1) only allow AMT workers that have a pass rate over 90% to participate in our human evaluation, which means the workers have a good record in AMT tasks; 2) pay the workers with a higher rate than similar tasks; 3) set verifications to make sure the human-generated question and answers are longer than 5 words, and filtering out bad question/answers that are clear copy/paste.

The evaluation result shows human evaluators think LTU answers are better than human-generated answers (crafted by another independent human subject after listening to the audio), which may be quite surprising. In our observation, LTU answers are usually more complete and detailed than human answers, which is the reason why they are rated better. We want to highlight that consistency and endurance are advantages of LTU as a machine. Below is an example:

> **Question:** "What type of sound is it?"
> **LTU Answer:** "The audio clip contains the sounds of a train, clickety-clack, and steam whistle blowing at a high pitch."
> **Human Answer:** "It was train moving sound."

### I.2 GPT-ASSISTED EVALUATION

As a supplement to human subjective evaluation, to quantitatively evaluate how well `LTU` follows the instruction, we follow (Peng et al., 2023a) to evaluate it with GPT assistance. Specifically, we first use GPT-3.5-Turbo and GPT-4 (GPT-4 is not used in generating the training data) to generate 1,000 new questions based on unseen audios sampled from the evaluation split of the strongly-labeled AudioSet using AIG as described in Section 3.2, respectively. We then input these questions

Figure 2: Screenshot of the human subjective evaluation questionnaire.

Table 14: The instruction following rate automatically rated by GPT-4.

| Model | Question Generated by | |
|---|---|---|
| | GPT-3.5-Turbo | GPT-4 |
| LTU | 95.9% | 96.9% |
| No Open-Ended Training | 26.9% | 22.5% |

and corresponding audios to `LTU` and let it produce the answer. Finally, we use GPT-4 to evaluate if the `LTU` output answers the question by the prompt "Below is a pair of question and response. Identify if the response answers the question. Return yes or no." As shown in Table 14, `LTU` has an instruction following rate over 95% for questions generated by both GPT-3.5-Turbo and GPT-4. In contrast, the model trained with only closed-ended tasks cannot follow instructions well (22.5%).

## J    EXISTING CAPTIONING BENCHMARK ISSUE IN MEASURING LTU

Table 15: Two sample LTU outputs that are semantically correct but get 0 SPICE scores due to not hitting the keywords in the 5 ground truths of each audio.

**Ground Truths:** drilling noise loud and continue. a loud machine running. a power tool sanding. a tool buzzing. a machine motor buzzing and humming.
**LTU Output:** a drill is running and vibrating. **SPICE score:** 0.0

**Ground Truths:** a muffled helicopter engine operating as paper crinkles in the background. aircraft engine running constantly. an engine running. an engine running consistently. airplane engine idles continuously.
**LTU Output:** a propeller is whirring loudly. **SPICE score:** 0.0

Existing audio captioning metrics are not optimal for models trained on diverse datasets with large vocabulary like LTU as they do not count synonyms as correct. We show two sample LTU outputs that are semantically correct but get 0 SPICE scores due to not hitting the keywords in the 5 ground truths of each audio. For this reason, we mainly use classification benchmarks in the ablation studies.

## K    ACOUSTIC FEATURE LEARNING AND DECOMPOSITION

Among our closed-ended tasks, one task is asking the model about the acoustic features. This task aims to train the model to recognize low-level features for better generalization. Original audio datasets do not have this information. We thus generate the acoustic feature description using GPT-3.5-Turbo with the prompt "describe the acoustic characteristic of {sound class name} sound precisely with a sentence less than 10 words". We generate 10 different descriptions for each sound class. The question for this task is "classify the sound events in the audio clip based on acoustic features" and its GPT-assisted paraphrases. The answer is a list of GPT-3.5-Turbo-generated acoustic features and the sound class names.

Table 16: Sample acoustic features generated from audio labels with GPT assistance. Most generated acoustic features are low-level descriptors and appear in pairs. We show the most frequent acoustic features generated based on the 527-class AudioSet labels, frequency shown in parentheses, e.g., "high pitched" appears in the descriptions of 243 audio classes, while its opposite concept "low pitched" is in the descriptions of 77 classes.

high pitched(243)-low pitched(77), loud(150)-soft(31), bright(99)-dull(10)
sharp(197)-mellow(44), deep(81)-shrill(43), metallic(72)-warm(49)
harsh(60)-soothing(20), clear(33)-muffled(22), intense(35)-gentle(17)

As shown in Table 16, most generated acoustic features are indeed low-level descriptors. More importantly, we find acoustic concepts usually appear in pairs, e.g., "high pitched" appears in the descriptions of 243 audio classes, while its opposite concept "low pitched" is in the descriptions of 77 classes. This allows LTU to learn better about the features.

To further check if LTU indeed learns acoustic concepts rather than just simply associates the acoustic concept with a sound class, we conduct a probe test. In our training data, samples of the "ambulance siren" class are always described as "high pitched". To check if LTU can disentangle the concept of a high-pitch and sound class of ambulance sirens. We manually lower the pitch (`librosa.pitch_shift`) of 53 evaluation audios of the "ambulance siren" class and check LTU's output to the question "What is the pitch?" on these audios. As shown in Figure 2, LTU's prediction aligns well with the actual pitch, indicating it indeed learns the concept of pitch rather than just associating it with a specific sound class.

Figure 2: To check if LTU can disentangle the concept of high-pitch and ambulance sirens. We manually lower the pitch (`librosa.pitch_shift`) of 53 evaluation audios of the "ambulance siren" class and check LTU's output to the question "What is the pitch?" on these audios. LTU's prediction aligns well with the actual pitch.

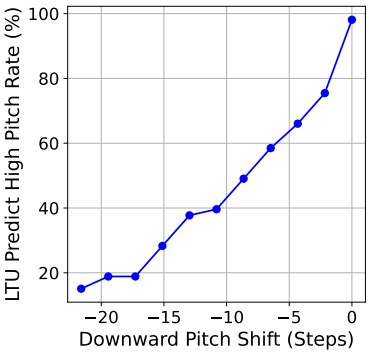

Table 17: Sample GPT-3.5-Turbo generated acoustic descriptions for 10 majority sound classes and 10 minority sound classes in AudioSet. For each sound class, we generate 10 different descriptions and show only one sample for each due to space limitations.

| Sound Class | # Samples in AudioSet | Sample GPT-3.5-Turbo Generated Acoustic Description |
|---|---|---|
| Music | 1,011,305 | distinctly pitched and timed |
| Speech | 1,010,480 | rich in frequency modulation |
| Vehicle | 128,051 | low, rumbling and loud |
| Inside, small room | 76,694 | intimate and reverberant |
| Guitar | 51,597 | bright and well-rounded |
| Plucked string instrument | 44,565 | plucked, staccato articulation of a string |
| Singing | 42,493 | vibrant and melodic |
| Animal | 40,758 | usually high-pitched and short |
| Electronic music | 38,958 | synthesized and exaggerated |
| Outside, rural or natural | 35,731 | generally calm and mellow |
| Dental drill, dentist's drill | 182 | sharp and metallic |
| Crushing | 176 | gritty and harsh |
| Hoot | 169 | high pitched and shrill |
| Finger snapping | 164 | percussive and sharp |
| Squawk | 160 | sharp and piercing |
| Splinter | 153 | sharp, loud and brief |
| Pulleys | 152 | a high-pitched, metallic ringing |
| Creak | 149 | high pitched, harsh and sharp |
| Gargling | 137 | low pitch, gurgling and wet |
| Toothbrush | 127 | high-pitched and buzzing |

### K.1    SAMPLE ACOUSTIC FEATURES

We present GPT-3.5-Turbo generated acoustic descriptions for 10 majority sound classes and 10 minority sound classes in AudioSet in Table 17. In general, these feature descriptions are factually correct and informative for both common and rare sounds.

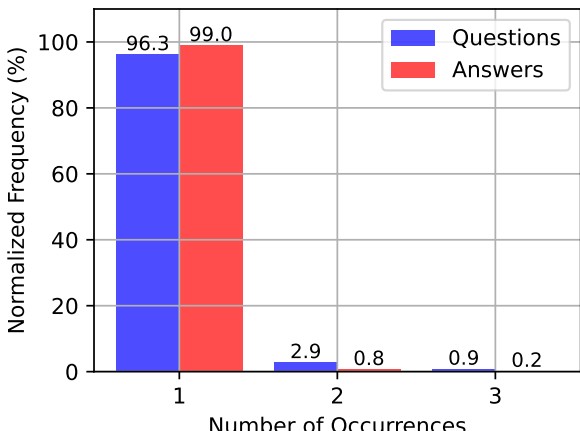

Figure 3: Histogram of question and answer occurrences in the 3.7M open-ended `OpenAQA` dataset.

## L  TRAINING DATA DIVERSITY

The proposed `OpenAQA` is very diverse. In Figure 3 we show the histogram of question and answer occurrences in the 3.7M open-ended `OpenAQA` dataset. Over 95% of questions and answers appear only once in the dataset, demonstrating the diversity of the dataset.

## M  FULL GPT-3.5-TURBO PROMPT AND SAMPLE OUTPUTS

In Table 2, we show the prompt for GPT-3.5-Turbo for generating AQAs. Due to space limitations, we remove some formatting instructions. Below is the full prompt we have used:

"Based on the following audio clip, generate 10 different types of complex open-ended questions that require step-by-step thinking, and corresponding step-by-step answers. The following information is provided: the sound events appear in the audio clip, together with its acoustic features, and corresponding onset and offset time stamps. A description of the content of the audio clip is also provided. Questions should be about the audio, e.g., which sound event is recognized and why (e.g., based on its acoustic feature), what can be inferred based on the combination of sound events; the temporal relationship between the sound events and what can be inferred from that; the potential scenario that such an audio clip could happen, if the audio clip is special (e.g., urgent, funny, interesting, abnormal, unique, etc) and why, what mood or atmosphere this audio clip conveys, etc. The more complex and diverse the question, the better. Format each QA pair in a single line as a JSON dictionary (key "q" for question, and "a" for answer, wrapped with { and }). Do not include any other explanation."

The output of GPT-3.5-Turbo is in the form of a JSON dictionary, as shown in the following example:

```
[{
  "q": "How would you describe the tone of the sound of the
       accelerating engine?",
  "a": "The tone of the sound of the accelerating engine is
       high-pitched, short and intense."
},
...
{
  "q": "What is the acoustic feature that distinguishes the
       sound of the ambulance siren from the generic impact sounds?",
  "a": "The acoustic feature that distinguishes the sound of the
       ambulance siren from the sound of generic impact sounds is that
       the former is high-pitched and wailing, while the latter is
       loud and sharp."
```

```
}]
```

# N   LOSS CURVE

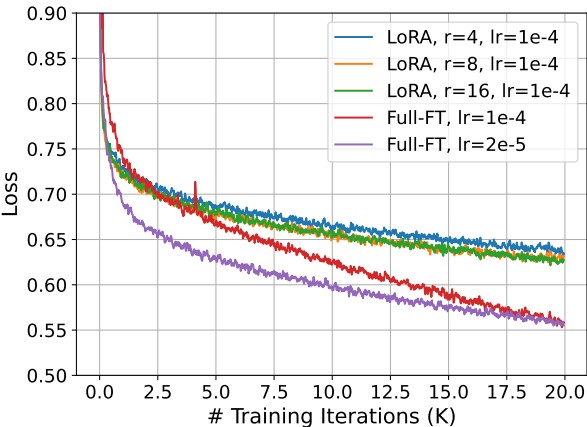

Figure 4: The Stage 3 training loss curve of various training settings.

We show the training loss curve of various training settings of Training Stage 3 (the last training) in Figure 4. While full-finetuned models (i.e., set LLaMA trainable) achieve noticeably lower losses compared to LoRA models, as shown in Table 5, they only perform slightly better on audio tasks. In addition, they dramatically lose the original language reasoning ability.

# O   MORE LTU DEMO SAMPLES

Due to space limitations, we are only able to present a limited number of demos. In Table 18, we show additional demos of `LTU`.

# P   DATASET DETAILS

## P.1   TRAINING DATASETS

As we mentioned in the main manuscript, we relabel existing public datasets including AudioSet (including a 500K subset of the original 2M weakly-labeled release (Gemmeke et al., 2017) and the 100K subset with temporally-strong labels (Hershey et al., 2021)), VGGSound (Chen et al., 2020a), FSD50K (Fonseca et al., 2021), AudioCaps (Kim et al., 2019), Freesound (Font et al., 2013), Clotho v2 (Lipping et al., 2019), and Sound Bible (soundbible.com, 2006) as our training data. For all these datasets, we only include data marked as training and validation samples and exclude any data marked as test or evaluation. A total of 845K unique audio clips are used. In the main manuscript, we discuss the relabeling method (the novel method) in detail but just briefly mention the source datasets, below we introduce them and our preprocessing method in detail.

**AudioSet-2M** (Gemmeke et al., 2017)

AudioSet-2M is the original release of AudioSet, it is a collection of 2 million 10-second audio clips excised from YouTube videos and labeled with the sounds that the clip contains from a set of 527 labels. AudioSet-2M is a *weakly labeled* and multi-label dataset, i.e., labels are given to a clip with no indication of the onset and offset timestamps, and every clip may have multiple labels.

While we would like to include more audio, from the audio source diversity consideration, we do not want audio from a single source to dominate the `OpenAQA` dataset. Therefore, we sample a 500K subset from AudioSet-2M. Specifically, we sample the audio using the following algorithm. First, we calculate the weight of each sound class $k$ as $w_k = 1/$number of samples labeled as the class, i.e., uncommon sound classes have a higher weight. Then, we calculate the weight of each audio

Table 18: More LTU Demo Samples. Answers are directly taken from the model output without modification. Notes in grey square brackets are not input to the model. Audios can be found in the supplementary materials.

---

**Sample 1. Bell Ringing Sound**

Question: What did you hear?    [Closed-Ended Question]
Answer: Music; Bell; Mechanisms; Generic impact sounds; Ding-dong.
Question: How many times the bell is rung?    [Follow-up Question, Detailed Perception Question]
Answer: The bell is rung three times.
Question: Describe the sound of the bell in this audio clip in detail.    [Follow-up Question, Detailed Perception Question]
Answer: The bell in this audio clip has a bright and ringing sound with a high pitch and short duration. It is a single ringing sound that is repeated several times in quick succession, creating a rhythmic pattern.
Question: What type is the bell?    [LTU knows what it does not know.]
Answer: The audio clip does not provide enough information to determine the type of the bell. It could be a church bell, a school bell, or any other type of bell that produces a similar sound.

---

**Sample 2. Gun Shot Sound**

Question: Describe the audio clip with a sentence.    [Closed-Ended Question]
Answer: Gunshots and footsteps are heard, followed by a man speaking and a beep.
Question: What is the possible scenario of this sound?    [Scene Understanding]
Answer: It is possible that the man is walking through a war zone or a dangerous area.
Question: What mood or atmosphere does this audio clip convey? Explain Why.    [Audio Mood Understanding]
Answer: This audio clip conveys a tense and potentially dangerous atmosphere, as the gunshots and the man's speech suggest that there is a potential threat or conflict.
Question: If I hear the sound, what should I do?    [Action Suggestion]
Answer: If you hear the sound, you should immediately seek shelter and call for help, as it could be a sign of danger or an emergency situation.
Question: If I hear the sound when I am with kids, what should I do?    [Action Suggestion]
Answer: If I hear the sound when I am with kids, I should immediately take them to a safe place and call the police or emergency services for assistance.

---

**Sample 3. Owls Night Calls**

Question: What sound is heard?    [Closed-Ended Question]
Answer: The sound of wind is heard throughout the entire clip, while the sound of insects, hoot, and owl are heard intermittently and at different time stamps.
Question: What insect is heard?    [Follow-up Question, Detailed Perception Question]
Answer: The insects heard in the audio clip are cricket.
Question: What insect is heard? Why?    [Prediction Explanation]
Answer: The insect heard is a cricket. It is high-pitched and chirpy, which is why it is recognized.
Question: Is this audio more likely to be recorded in the morning or evening? Why?    [Scene Understanding]
Answer: This audio is more likely to be recorded in the evening because the owl hooting sound is more prominent during the night.
Question: What can be inferred from the audio? Think step by step.    [Chain-of-Thought Reasoning]
Answer: The audio clip contains the sound of wind, insects, and hooting. The wind is blowing and the insects are chirping. The hooting sound is likely from an owl or other bird of prey. The sound of the wind is high-pitched and hissing, while the insects are high-pitched and chirpy. The hooting sound is deep and resonant, suggesting a larger bird of prey or owl. The combination of these sounds suggests a natural outdoor environment, possibly at night. The hooting sound could also suggest a sense of danger or urgency, as owls are known to be predators.
Question: What type of video can this audio be used for dubbing?    [Audio Usage Suggestion]
Answer: This audio can be used for dubbing a horror movie.

---

**Sample 4: Busy Signal Sounds in a Phone Call**

Question: Classify the sound events in the audio clip.    [Closed-Ended Question]
Answer: Busy signal; Male speech, man speaking; Background noise; Tick; Speech
Question: Classify the sound events in the audio clip based on acoustic features.    [Closed-Ended Question]
Answer: High-pitched and metallic ->Telephone bell ringing; Aperiodic, temporally varying ->Speech
Question: Write an audio caption describing the sound.    [Closed-Ended Question]
Answer: A busy signal and a man speaking is heard, with background noise in the background.
Question: What makes you think it is a busy signal?    [Reasoning Question]
Answer: The busy signal is characterized by a continuous tone, which is different from the sound of a ringing phone, which is a series of short pulses. The continuous tone suggests that the line is busy and cannot be reached, while the ringing tone suggests that the line is available but no one is answering.
Question: What should the man do?    [Action Suggestion]
Answer: The man should wait for the busy signal to end before trying to call again or try a different phone number if necessary to reach the desired person or organization. Alternatively, the man could try to contact the person or organization through other means, such as email or text messaging, if the phone call is not urgent or time-sensitive.

---

**Sample 5: Classify Rooster and Hen Sound.**

Question: Is the sound Rooster or Hen?    [Closed-Ended Question]
Answer: The sound is Rooster.
Question: No, the sound is actually from a hen. Can you tell me why you are wrong?    [Explain the Wrong Prediction]
Answer: I apologize for my mistake earlier. The sound in this audio clip is indeed from a hen, not a rooster. This can be inferred based on its lower pitch and shorter duration compared to a rooster's crow.

---

sample $a$ as $w_a = \sum_{k \in K} w_k$, where $K$ is the labels associated with audio $a$, i.e., audio samples containing more and uncommon sound events have a higher weight. Finally, we select the top 500K audio samples that have higher weights. Due to the constant change in Youtube video availability (e.g., videos being removed, or taken down), there is a natural shrinkage from the original dataset, so the 500K samples are selected from the 1,772,023 audio clips we were able to download.

**Temporally-Strongly Labeled AudioSet** (Hershey et al., 2021)

Temporally-Strongly Labeled AudioSet is a subset of the original AudioSet-2M with additional annotations. Specifically, the start and end times of each event are also annotated. It consists of 934,821 sound events across the 103,463 10-second audio clips. The label set of temporally-strongly labeled AudioSet is slightly different from the original AudioSet label set (the label set of temporally-strongly labeled AudioSet contains 447 labels, of which 376 are shared with the original 527-class label set).

Since temporally-strongly labeled AudioSet contains rich annotations, we include the entire training set in `OpenAQA`. Similar to the original AudioSet, there is a natural shrinkage due to some of the Youtube videos being unavailable. We use 101,791 audio clips that we were able to download (among them, 91,075 clips are captioned by the WavCaps project (Mei et al., 2023)).

**VGGSound** (Chen et al., 2020a) VGGSound is a collection of 200K 10-second audio clips excised from YouTube videos annotated with 309 classes. Different from AudioSet, each VGGSound audio sample contains only one audio event. We use 183,727 training audio clips that we were able to download.

**FSD50K** (Fonseca et al., 2021) FSD50K contains 37,134 audio clips for training and 4,170 audio clips for validation from the Freesound project (Font et al., 2013). The audio clips are unequally distributed in 200 sound classes drawn from the AudioSet ontology. We use both the training and validation set, a total of 41,304 audio samples to train `LTU`.

**AudioCaps** (Kim et al., 2019) AudioCaps is a subset of AudioSet with additional human-written audio caption annotations collected via crowdsourcing. The training and validation set contains 49,838 and 495 audio clips, respectively. Each training audio clip is annotated with 1 caption and each validation audio clip is annotated with 5 captions. We use both the training and validation set, a total of 46,462 audio clips and 48,298 captions that we were able to download.

**Freesound**(Font et al., 2013) Freesound is an online collaborative sound-sharing project launched in 2005 with more than 560K audio clips. In this paper, we use 91,434 audio clips that are shorter than 10 seconds and captioned by the WavCaps project (Mei et al., 2023).

**Clotho V2** (Lipping et al., 2019) Clotho is an audio caption dataset with audio sampled from Freesound and captions collected by Amazon Mechanical Turk. Each audio is annotated with 5 captions. We use 3,741 audio clips from the development set and 1045 audio clips from the validation set to train `LTU`.

**Sound Bible** (soundbible.com, 2006) SoundBible is a website sharing sound effects and audio clips. We use 1,225 audio clips that are captioned by the WavCaps project (Mei et al., 2023).

## P.2 EVALUATION DATASETS AND PROTOCOL

### P.2.1 BEST SUPERVISED AND SPECIALIZED MODELS IN TABLE 4

Due to the space limitation, we only show the results of the best supervised and specialized models in Table 4. These models are Chen et al. (2022) (ESC-50), Kong et al. (2020) (DCASE), Elizalde et al. (2023) (VocalSound and TUT), Cramer et al. (2019) (Beijing Opera), Gong et al. (2022a) (VGGSound), Gong et al. (2021a) (FSD50K), Huang et al. (2022) (AudioSet), Kim et al. (2022) (AudioCaps), and Mei et al. (2022) (Clotho V2).

### P.2.2 ZERO-SHOT EVALUATION

**VocalSound** (Gong et al., 2022b) VocalSound is a dataset consisting of 21,024 crowdsourced recordings of laughter, sighs, coughs, throat clearing, sneezes, and sniffs from 3,365 unique subjects. We evaluate `LTU` on the VocalSound evaluation set consisting of 3,594 audio clips and report the 6-class

classification top-1 accuracy. Note that VocalSound is completely independent of the `LTU` training data, therefore it is a zero-shot evaluation.

**TUT 2017** (Mesaros et al., 2016) TUT 2017 is a dataset consisting of 10-second audio segments from 15 acoustic scenes recorded from distinct locations. In the evaluation set, each of the 15 acoustic scenes has 108 segments and the total number of evaluation samples is 1,620. We evaluate `LTU` on the evaluation set of TUT 2017 and report the top-1 accuracy. Note that TUT 2017 is completely independent of the `LTU` training data, therefore it is a zero-shot evaluation.

**Beijing Opera** (Tian et al., 2014) Beijing Opera is an instrument classification dataset comprising 4 percussion instruments: "bangu": "clapper-drum", "naobo": "cymbals", "daluo": "large gong", and "xiaoluo": "small gong". We use "clapper-drum", "cymbals", "large gong", and "small gong" as the name of labels. We evaluate `LTU` on the entire 236 Beijing Opera audio samples and report the top-1 accuracy. Note that Beijing Opera is completely independent of the `LTU` training data, therefore it is a zero-shot evaluation.

### P.2.3 WEAK ZERO-SHOT EVALUATION

**ESC-50** (Piczak, 2015) The ESC-50 dataset consists of 2,000 5-second environmental audio recordings organized into 50 classes. The standard evaluation protocol is 5-fold cross-validation, i.e., using 1,600 samples to train the model and using the rest 400 samples to evaluate the model. We do not do any training on ESC-50, instead, we directly evaluate `LTU` on the entire 2,000 audio clips and report the top-1 accuracy. Note that though we do not mix ESC-50 in our training set, ESC-50 is sampled from the Freesound project, which is also the source of part of our training data, we, therefore, call this a weak zero-shot setting.

**DCASE2017 Task 4** (Mesaros et al., 2017) DCASE 2017 Task 4 consists of audio clips of 17 sound events divided into two categories: "Warning" and "Vehicle". We evaluate `LTU` on the evaluation set of DCASE 2017 Task 4 consisting of 1,350 audio samples and report the micro F1-score. Note that though we do not mix DCASE 2017 Task 4 in our training set, it is sampled from the AudioSet project, which is also the source of part of our training data, we, therefore, call this a weak zero-shot setting.

### P.2.4 IN-DOMAIN EVALUATION

Even for a dataset whose training split has been used to train `LTU`, due to the multi-dataset training setting and the free-form output nature of `LTU`, the prediction search space of `LTU` is still much larger than conventional models trained solely on a single dataset. In addition, `LTU` does not use any dataset-specific training tricks (Gong et al., 2021a; Moore et al., 2023). Therefore, it is not exactly fair to compare `LTU` with dataset-specialized models.

**VGGSound** (Chen et al., 2020a) The training split of VGGSound is part of the `LTU` training data. We evaluate `LTU` on the VGGSound evaluation set consisting of 15,446 audio samples and report the top-1 accuracy.

**FSD50K** (Fonseca et al., 2021) The training and validation split of FSD50K is part of the `LTU` training data. We evaluate `LTU` on the FSD50K evaluation set consisting of 10,231 audio samples and report the mean average precision (mAP).

**AudioSet** (Gemmeke et al., 2017) The training split of AudioSet is part of the `LTU` training data. We evaluate `LTU` on the AudioSet evaluation set consisting of 17,249 audio samples and report the mean average precision (mAP).

**AudioCaps** (Kim et al., 2019) The training and validation split of AudioCaps is part of the `LTU` training data. We evaluate `LTU` on the AudioCaps evaluation set consisting of 901 audio clips and 4,505 captions and report the SPICE score. It is worth noting that though AudioCaps are annotated by crowd workers (humans), the AudioSet labels are used as "word hints" to the worker in the annotation process, so the vocabulary of the captions may be relatively limited. This means a model trained solely or mainly on the dataset may fit better with its vocabulary and get a higher score on the evaluation set while a general model that has a larger vocabulary may have a lower score because the evaluation metrics like SPICE and CIDER view synonyms as a wrong prediction.

**Clotho V2** (Drossos et al., 2020) The development and validation split of Clotho is part of the `LTU` training data. We evaluate `LTU` on the Clotho evaluation set consisting of 1,045 audio clips and 5,225 captions and report the SPICE score. Compared with AudioCaps (Kim et al., 2019), Clotho does not provide ground truth labels as "word hints" to the annotators, and more rigorous screening and sanitation are conducted. For example, text format is manually corrected for consistency (e.g., replacing "it's" with "it is"), unique words are rephrased to make sure they appear in both training and evaluation sets, the length of the caption is controlled to be between 8 to 20, etc. Thus model trained solely or mainly on the dataset may fit better with the vocabulary and style and get a higher score on the evaluation set while a general model that has a larger vocabulary and more free-form output may have a lower score because the evaluation metrics like SPICE and CIDER view synonyms as wrong prediction, and its output may be too brief (less than 8 words) or too specific (over 20 words) compared with the ground truth caption.

