# OpenReview forum: "Listen, Think, and Understand"
_ICLR.cc/2024/Conference — ICLR 2024 poster_

### Official Review · Reviewer_wB1w · 2023-10-28

**Soundness:** 3 good
**Presentation:** 3 good
**Contribution:** 3 good
**Rating:** 8
**Confidence:** 4

**Summary:**

The authors propose to connect an audio encoder (AST) with a large language model (LLM) through LoRA adapter tuning for audio understanding. To fine-tune this model, the authors also propose and curate an OpenAQA-5M dataset which mixes of existing audio tasks such as audio classification, audio captioning, and sound event detection as close-ended tasks, and leverage LLMs to generate question and answer pairs given text metadata. The authors further conducted human evaluation to verify these generated data.

**Strengths:**

- OpenAQA-5M is a good contribution to provide open-ended question answering in audio domain, especially it is verified with human evaluation.
- Ablation study shown in table 5 provides good insights for choices of LoRA params and the benefit of curriculum in staged training.

**Weaknesses:**

- For the open-ended questions, this work seems to focus mainly on solely LLMs assisted question answer generation. It is not intuitive to understand to what extent the model relies on the input audio versus on the common sense knowledge that is already encoded in the LLMs. It would be great to define and identify beyond current close-ended tasks with new lower level tasks which really require using the audio, such as counting sound events, ordering of events, etc. These type of questions might already exist in the proposed dataset, it would provide more insights to dive deeper into those.
- Table 5 on the right for the training curriculum, it would be great to also include the language instruction following rate. And further discuss the correlation between the classification performance and the instruction following rate, if there is any insights that can be drawn.

**Questions:**

- The acoustic features mentioned in 3.1 are also generated from LLMs, how are these verified?
- In the temporal analysis paragraph in 3.1, how are the understanding of order of sounds evaluated?
- In 5.2.1, LTU can answer follow-up questions about the details, there is an example where LTU shows that the bell rings three times, are there other examples showing the counting capabilities? Is it possible to further quantify this?
- In 5.2.1, multi-turn conversation is mentioned, are the previous audio tokens and questions also passed in as context or are the multi-turn independent each turn? This can be another interesting property for few-shot in-context learning if possible.

---

> ### Author Response · Authors · 2023-11-22
> **Author Response to Reviewer wB1w (1/4) Temporal Analysis Experiments**
>
> Dear Reviewer wB1w,
>
> Thank you for the positive feedback and all these very valuable and constructive questions/suggestions! Let us respond to your questions point by point.
>
> ---
> **Question 1 - Additional Audio-Conditioned Experiments**
>
> > For the open-ended questions, this work seems to focus mainly on solely LLMs assisted question answer generation. It is not intuitive to understand to what extent the model relies on the input audio versus on the common sense knowledge that is already encoded in the LLMs. It would be great to define and identify beyond current close-ended tasks with new lower level tasks which really require using the audio, such as counting sound events, ordering of events, etc. These type of questions might already exist in the proposed dataset, it would provide more insights to dive deeper into those.
>
> We totally agree with the reviewer on this point! And this is the reason why we rigorously evaluate the model performance on closed-ended tasks while the main novelty on open-ended tasks. In the paper, Appendix K, we conduct a probe experiment to check if LTU can really understand the sound concepts rather than associates the acoustic concept with a sound class. In our training data, samples of the ``ambulance siren`` class are always described as ``high pitched``. To check if LTU can disentangle the concept of a high-pitch and sound class of ambulance sirens. We manually lower the pitch (with librosa.pitch_shift) of 53 evaluation audios of the ambulance siren class and check LTU's output to the question ``What is the pitch?`` on these audios. As shown in Figure 2 of the paper, LTU's prediction aligns well with the actual pitch, indicating it indeed learns the concept of pitch rather than just associating it with a specific sound class.
>
> ---
> **Question 2 - Temporal Analysis  Experiments**
> > It would be great to define and identify beyond current close-ended tasks with new lower level tasks which really require using the audio, such as counting sound events, ordering of events, etc.
>
> >In the temporal analysis paragraph in 3.1, how are the understanding of order of sounds evaluated?
>
> >In 5.2.1, LTU can answer follow-up questions about the details, there is an example where LTU shows that the bell rings three times, are there other examples showing the counting capabilities? Is it possible to further quantify this?
>
> We sincerely appreciate the reviewer's suggestion to explore the temporal analysis capability of the model, and in response, we are proceeding with the following new experiments. Specifically, we conduct experiments on the audio event order recognition task (seen in the closed-ended training stage) and audio event counting task (unseen in the closed-ended training stage) with ESC-50, a dataset not being used in training. For both experiments, we only include the cases in which LTU follows the instruction.
>
> ---
> **1. Recognizing the Order of Sound Events**
>
> We create a test set where each test sample consists of two audio samples from different categories of the ESC50 dataset, with no temporal overlapping. We then ask LTU to predict the order of the sound events with the prompt ``Which sound begins and ends first?``. We use regular expressions and cosine similarity (based on gpt-text-embedding-ada) to interpret the raw LTU output. For example, for the LTU raw output ``the applause starts first, while the footsteps follow in a rhythmic pattern afterward.``, we first extract ``applause`` and ``footsteps follow in a rhythmic pattern`` using regular expressions. Then we compare the cosine similarity between the text embedding of the ground truth ``clapping`` with the text embeddings of ``applause`` and ``footsteps follow in a rhythmic pattern``, respectively. Since the cosine similarity between the text embedding of the ground truth ``clapping`` and the LTU prediction ``applause`` is higher, we count this sample as correct.
>
> We evaluate LTU on 550 such samples, obtaining 390 correct and 160 incorrect answers, achieving an accuracy of 70.9%. This shows that LTU can recognizes the order of sound events with a reasonable accuracy.

---

> > ### Author Response · Authors · 2023-11-22
> > **Author Response to Reviewer wB1w (2/4) Temporal Analysis Experiments**
> >
> > ---
> > **Question 2 - Temporal Analysis Experiments** (cont.)
> >
> > ---
> > **2. Counting the Number of Appearances of Sound Events**
> >
> > In Table 6, the first sample shows a demo where LTU recognizes the bell rings *three times*. In this section, we further quantify LTU's capacity in counting the number of appearances of sound events. Note that this task is not in our closed-ended training. One problem in evaluating this task is that the definition of the number of appearances is not clear for all sound classes, e.g., it is hard to define what an appearance of a train sound is as it could consist of multiple click-clack sounds. Thus, we create a test set where each test sample consists of one or two (repeated) audio samples from the ESC50 dataset, with no temporal overlapping. We then ask LTU to count the sounds in the audio clips using the prompt, ``In how many instances is the sound of {:s} heard?``. Since one ESC50 audio sample may contain sounds multiple times (e.g., three dog barking sounds), we calculate the Pearson Correlation Coefficient (PCC) between the LTU output count and the number of ESC50 audio clips in the test sample.
> >
> > | Sound Class    | Pearson Correlation Coefficient (PCC) | LTU Output Count Ratio for Double vs. Single ESC50 Audio Samples (Ideal = 2) |
> > |----------------|:-------------------------------------:|:----------------------------------------------------------------------------:|
> > | Chainsaw       |                  0.81                 |                                      2.5                                     |
> > | Brushing Teeth |                  0.79                 |                                      2.6                                     |
> > | Helicopter     |                  0.77                 |                                      2.1                                     |
> > | Siren          |                  0.58                 |                                      2.0                                     |
> > |                |                  ...                  |                                                                              |
> > | Train          |                  0.02                 |                                     1.01                                     |
> > | Toilet Flush   |                 -0.03                 |                                     0.98                                     |
> > | Frog           |                 -0.18                 |                                     0.92                                     |
> >
> > As shown in the above Table, we find a high correlation for many sound classes. Additionally, we calculate the ratio between LTU output counts for test samples consisting of two ESC50 samples and test samples consisting of one ESC50 sample. Ideally, this value should be 2 because the number of appearances of sounds in two ESC50 samples should be twice the number of appearances of sounds in one ESC50 sample. We find it is indeed close to 2 for many sound classes. However, there are some sound classes that LTU cannot count well, potentially due to an undefined number of sound appearances for these classes. This demonstrates the sound event counting capability of LTU, although it has not been explicitly trained for this task in the closed-ended training stage.
> >
> > ---
> > **Summary**
> >
> > To summarize, LTU demonstrates reasonable ability in temporal analysis. However, compared with classification and captioning tasks, its temporal analysis ability is weaker. We hypothesize that this is due to two reasons:
> > - First, we aggressively pool the audio tokens to save computation, which may hurt fine-grained tasks.
> > - Second, LTU has not been trained with a sufficient amount of data for temporal analysis, as reported in [1], such training is crucial for fine-grained tasks.
> >
> > [1] Peng, Zhiliang, et al. "Kosmos-2: Grounding Multimodal Large Language Models to the World." 2023.
> >
> > ---
> >
> > We thank the reviewer again for the very valuable comment. In the revised paper, we have added this discussion to **Appendix H LTU Temporal Analysis Experiments**. We hope the new section will provide readers with a deeper understanding of LTU's performance in temporal analysis and its limitations. Furthermore, we hope that these experiments can partly answer the question "To what extent the model rely on the input audio versus on the common sense knowledge that is already encoded in the LLMs?"

---

> > > ### Author Response · Authors · 2023-11-22
> > > **Author Response to Reviewer wB1w (3/4) Acoustic Feature Description**
> > >
> > > ---
> > > **Question 3 - Acoustic Feature Description Verification**
> > >
> > > >The acoustic features mentioned in 3.1 are also generated from LLMs, how are these verified?
> > >
> > > We thank the reviewer for the question. We utilize GPT-3.5-Turbo, a strong large language model, to generate acoustic feature descriptions. These descriptions are used to train the LTU model to enhance its capability to recognize acoustic features alongside sound classes and to enable LTU to generalize predictions for unseen sound classes based on their acoustic features. In the following table, we show examples of GPT-3.5-Turbo generated descriptions for 10 major and 10 minor sound classes from AudioSet. For diversity consideration, for each sound class, we generate 10 different descriptions but show only one sample in the Table for each due to space limitations. Our manual evaluation confirms that these feature descriptions are high-quality, being both factually accurate and informative for a range of common and rare sounds.
> > >
> > > | Sound Class                   | # Samples in AudioSet | Sample GPT-3.5-Turbo Generated Acoustic Description |
> > > |-------------------------------|:---------------------:|:---------------------------------------------------:|
> > > | Music                         |       1,011,305       |             distinctly pitched and timed            |
> > > | Speech                        |       1,010,480       |             rich in frequency modulation            |
> > > | Vehicle                       |        128,051        |                low, rumbling and loud               |
> > > | Inside, small room            |         76,694        |               intimate and reverberant              |
> > > | Guitar                        |         51,597        |               bright and well-rounded               |
> > > | Plucked string instrument     |         44,565        |      plucked, staccato articulation of a string     |
> > > | Singing                       |         42,493        |                 vibrant and melodic                 |
> > > | Animal                        |         40,758        |            usually high-pitched and short           |
> > > | Electronic music              |         38,958        |             synthesized and exaggerated             |
> > > | Outside, rural or natural     |         35,731        |              generally calm and mellow              |
> > > | Dental drill, dentist's drill |          182          |                  sharp and metallic                 |
> > > | Crushing                      |          176          |                   gritty and harsh                  |
> > > | Hoot                          |          169          |               high pitched and shrill               |
> > > | Finger snapping               |          164          |                 percussive and sharp                |
> > > | Squawk                        |          160          |                  sharp and piercing                 |
> > > | Splinter                      |          153          |                sharp, loud and brief                |
> > > | Pulleys                       |          152          |           a high-pitched, metallic ringing          |
> > > | Creak                         |          149          |            high pitched, harsh and sharp            |
> > > | Gargling                      |          137          |             low pitch, gurgling and wet             |
> > > | Toothbrush                    |          127          |               high-pitched and buzzing              |
> > >
> > > One problem related to this is that we apply the same set of descriptions to *all* samples labeled of the sound class. While these descriptions are generally accurate for the majority of samples, there can be exceptions. For instance, the ``vehicle`` class is described as ``low, rumbling, and loud``, but this might not hold true for some samples with quieter vehicle sounds. However, we observed that most generated acoustic features appear in **contrasting pairs**. For example, ``high pitched`` is used in the descriptions of 243 audio classes, whereas its opposite, ``low pitched``, is found in 77 classes. This presence of opposing features allows the LTU model to effectively disentangle the specific acoustic feature from the sound class, even when we provide the same set of features for each sound class.
> > >
> > > | The most frequent acoustic features generated based on the 527-class AudioSet labels, frequency shown in parentheses |
> > > |:--------------------------------------------------------------------------------------------------------------------:|
> > > |                      high pitched(243)-low pitched(77), loud(150)-soft(31), bright(99)-dull(10)                      |
> > > |                           sharp(197)-mellow(44), deep(81)-shrill(43), metallic(72)-warm(49)                          |
> > > |                         harsh(60)-soothing(20), clear(33)-muffled(22), intense(35)-gentle(17)                        |

---

> > > > ### Author Response · Authors · 2023-11-22
> > > > **Author Response to Reviewer wB1w (4/4) Other Comments**
> > > >
> > > > ---
> > > > **Question 4 - Classification performance vs the instruction following rate**
> > > >
> > > > >Table 5 on the right for the training curriculum, it would be great to also include the language instruction following rate. And further discuss the correlation between the classification performance and the instruction following rate.
> > > >
> > > > We thank the reviewer for the suggestion. Following the reviewer's suggestion, we add the instruction following rate for Table 5 right. Please see the following Table:
> > > >
> > > > | Curriculum Setting                                 | Audio Classification Performance | Pure Language Task Instruction Following Rate |
> > > > |----------------------------------------------------|:--------------------------------:|:---------------------------------------------:|
> > > > | LTU (trained w/ default curriculum)                |               50.3               |                      87.6                     |
> > > > | No Curriculum (same total tr. iterations)          |               23.0               |                    **67.7**                   |
> > > > | No Open-ended Training (Stage 1&2&3 only)          |               37.1               |                      87.8                     |
> > > > | No Open-ended Training (same total tr. iterations) |               47.3               |                      86.6                     |
> > > > | Use a large learning rate of 1e-3 in Stage 4       |               49.5               |                    **75.2**                   |
> > > > | Add half an epoch to the training in Stage 4       |               50.3               |                      86.7                     |
> > > > | Train for only one epoch in Stages 1&2             |               49.7               |                      88.5                     |
> > > > | Cut tr. iterations by 50% across all stages        |               45.3               |                      87.9                     |
> > > >
> > > > Interestingly, the relationship between the audio classification task (reflecting the model's audio perception ability) and the pure language task instruction following rate (indicative of the model's original language ability) is not linear. We find that the **training setting** significantly impacts the model's language abilities. It's possible for two models with similar audio perception performance to exhibit markedly different capabilities in language tasks. There are two settings that can lead to a lower language task instruction following rate:
> > > >
> > > > - Absence of a training curriculum. This finding reinforces the necessity of a structured training curriculum for optimal model development.
> > > > - Using a too large learning rate. Though the audio perception performance does not change much, training a model with a learning rate of 1e-3 can diminish its performance in language tasks compared with a learning rate of 1e-4.
> > > >
> > > > We thank the reviewer once again for raising this insightful question. We will incorporate this finding into the revised version of our paper.
> > > >
> > > > ---
> > > > **Question 5 - Multi-turn conversation**
> > > > >In 5.2.1, multi-turn conversation is mentioned, are the previous audio tokens and questions also passed in as context or are the multi-turn independent each turn? This can be another interesting property for few-shot in-context learning if possible.
> > > >
> > > > We appreciate the reviewer's question. In Table 6, all questions are independent, i.e., they are asked without prior context. Our intention is to avoid providing the model with information that might hint at the answer. By 'follow-up question,' we refer to posing a new question that seeks further details.
> > > >
> > > > However, LTU model inherently supports multi-round dialogue. We have conducted preliminary experiments in this direction. For instance, in the ESC-50 classification, for samples that the LTU initially misclassified, we can prompt it to make a correction
> > > >
> > > > ``Input: Is the sound Rooster or Hen? Why?``
> > > >
> > > > ``LTU Output: It is a Rooster.``
> > > >
> > > > ``Input: No, the sound is actually from a hen. Can you tell me why you are wrong?``
> > > >
> > > > ``LTU Output: I apologize for my mistake earlier. The sound in this audio clip is indeed from a hen, not a rooster. This can be inferred based on its lower pitch and shorter duration compared to a rooster's crow.``
> > > >
> > > > Although this sample appears promising, we acknowledge that rigorously quantifying the success rate of such corrections, and examining the impact of model size and training on it, are non-trivial tasks. In the context of in-context learning, the design of the prompt is crucial. Considering the current paper is already quite long, we plan to address these aspects in a future paper.
> > > >
> > > > ---
> > > > We have also incorporated new experiments in response to other reviewers including scaling up the LLM and comparing with Pengi.
> > > >
> > > > We hope we addressed all your concerns, and in this case would be happy if the reviewer could kindly consider increasing the score, and if not we are more than willing and happy to engage in a discussion with the reviewer to answer further questions and/or provide more clarifications as needed.

---

> > > > > ### Comment · Reviewer_wB1w · 2023-11-22
> > > > >
> > > > > Thank the authors for your through responses to most of the questions. With the extra answers, I encourage the authors to include them in the paper where it is appropriate. I am increasing the rating.

---

> > > > > > ### Author Response · Authors · 2023-11-22
> > > > > > **Author response**
> > > > > >
> > > > > > Dear Reviewer wB1w
> > > > > >
> > > > > > We sincerely appreciate your prompt review of our response and increasing the score.
> > > > > >
> > > > > > Yes, we have added the new experiments to the updated paper, in Appendix E, F, J, H, and M. You should be able to download it by clicking the top-right PDF button of the page.
> > > > > >
> > > > > > To make it easier to track the new sections, we temporarily put all of them in the Appendix but will consider moving them to the main manuscript in the next version.
> > > > > >
> > > > > > Thanks again.

---

### Official Review · Reviewer_oFtB · 2023-10-30

**Soundness:** 3 good
**Presentation:** 3 good
**Contribution:** 3 good
**Rating:** 6
**Confidence:** 4

**Summary:**

This paper proposes an audio foundation model, Listen, Think, and Understand by combining an existing audio encoder and LLM. It also creates a new audio dataset called OpenAQA-5M for training the proposed LTU, so that it can enhance the existing audio perception capabilities and also provide a clear explanation of training details.

The paper effectively delivers that the existing audio models have limitations in reasoning and comprehensibility and suggests that combining audio models and LLM would resolve the problems that the existing audio techniques have suffered.

It is also well-supported experimentally in two key areas: audio perception and reasoning abilities. Audio perception is evaluated through tasks like classification and captioning. Reasoning ability is evaluated through human assessment, and the paper claims that it outperforms GPT-4.

**Strengths:**

1. This paper claims that it is the first time designing a reasoning comprehension-capable model.
2. Details of training and dataset are logical and delicate. It handles the hallucination problem of LLM by training close-ended dataset and then non-answerable question-answer pairs. It considers the training direction to be "first to perceive, and then comprehend the sound" so that the training starts from using close-ended datasets to open-ended datasets. The paper shows the reasonable claim that it is necessary to gradually train the model from close-ended datasets to open-ended ones because if the open-ended dataset is trained first, the model is heavily dependent on language capability so it is hard to train the audio representation.
3. The paper is clear and easy to understand.

**Weaknesses:**

1. There seems to be a lack of thought about model structure and loss. It is just a combination of the strong pretrained LLM and the existing audio encoder, AST. Utilizing strong pretrained LLM with multimodal inputs has an alignment issue. Thus, while concatenating the audio feature and the text feature can introduce desired performance, there could be some advancements not just combining pretrained audio model and LLM. For example, BLIP-2 [1] solves misalignment between text and image using 3 losses: 1) Image-Text matching, 2) Image-Grounded Text generation, and 3) Image Text Contrastive Learning. The simple combination of the audio model and LLM does not seem to be novel.
2. The concurrent works show higher performances. Compared to Pengi, the closed-ended audio task performances are lower. I understand that the proposed paper is focusing on the open-ended problem, but it would be better to elaborate more in detail that the proposed paper is competitive compared to the concurrent works.

[1] Li, Junnan, et al. "Blip-2: Bootstrapping language-image pre-training with frozen image encoders and large language models." arXiv preprint arXiv:2301.12597 (2023).

**Questions:**

1. I don't understand 64(time) X 16(frequency) in the 5th line of Audio Encoder part in Section 2: LTU Model Architecture. Is it typo(I think it should be 8)? Or what is the meaning of frequency 16?
2. I want to know how the dataset is formulated (Q and A) that is generated from GPT.

---

> ### Author Response · Authors · 2023-11-22
> **Author Response to Reviewer oFtB (1/5) Audio-Text Alignment**
>
> Dear Reviewer oFtB,
>
> Thank you so much for taking the time to read our paper and providing very valuable comments and suggestions regarding the model architecture and comparing LTU with Pengi. Please see the following for our point-by-point response. Please also note that we have added substantially more materials in the revised paper (please check the updated PDF file).
>
> ---
> **Question 1. Audio-Text Alignment**
>
> >There seems to be a lack of thought about model structure and loss. It is just a combination of the strong pretrained LLM and the existing audio encoder, AST. Utilizing strong pretrained LLM with multimodal inputs has an alignment issue. Thus, while concatenating the audio feature and the text feature can introduce desired performance, there could be some advancements not just combining pretrained audio model and LLM. For example, BLIP-2 [1] solves misalignment between text and image using 3 losses: 1) Image-Text matching, 2) Image-Grounded Text generation, and 3) Image Text Contrastive Learning. The simple combination of the audio model and LLM does not seem to be novel.
>
> We understand the reviewer’s concern about the architecture and are grateful for the highly constructive suggestion of adopting BLIP-2-like alignment training. Just as the reviewer points out, we connect the audio encoder with the large language model with only a linear layer and before LTU training, this audio encoder has not been trained for any audio-text alignment tasks.
>
> In our design, the AST's output serves as *soft prompts* for the large language model. According to [1,2], these (multimodal) *soft prompts* do not necessarily need to be in the text embedding space. In practice, some multi-modal large language models in the vision community such as PaLM-E, LLaVA, and Kosmos use a similar architecture and achieve good performance. One reason for not allowing us to directly use Q-Former in BLIP-2 is that the learnable queries are not guaranteed to be temporally aligned, which is important for audio tasks.
>
> We sincerely appreciate the reviewer's suggestion to explore an additional training stage for audio-text alignment, and in response, we are proceeding with the following new experiments.
>
> [1] Lester, Brian, et al. The Power of Scale for Parameter-Efficient Prompt Tuning. EMNLP 2021.
>
> [2] Driess, Danny, et al. PaLM-E: An Embodied Multimodal Language Model. 2023.
>
> In the new experiment, we add an additional audio-text alignment training stage before LTU training. We test two types of alignment modules: one is the linear layer used in the original LTU, and the other is a Transformer layer, which allows for more model capacity in the alignment. We train the alignment module on 1.6M audio-text pairs (comprising audio labels or captions) for 10 epochs, using a batch size of 1536 distributed over 4 GPUs, with 384 samples on each GPU. We start with an initial learning rate of 1e-3 and halve it after each epoch. During this stage, we keep the audio and LLaMA text encoders frozen and only train the alignment module. We use the following loss:
>
> $$
>  \mathcal{L} = \mathcal{L}_c + \lambda \cdot \mathcal{L}\_{mse}
> $$
>
> $$
> \mathcal{L}_{mse} = \frac{1}{N} \sum\_{i=1}^N (e^a_i - e^t_i)^2
> $$
>
> $$
> \mathcal{L}_\mathrm{c} = - \frac{1}{N} \sum\_{i=1}^N {\rm log}  \left[ \frac{ {\rm exp} (s\_{i,i}/\tau)}{\sum\_{k \neq i} {\rm exp} (s\_{i,k}/\tau) + {\rm exp} (s\_{i,i}/\tau)} \right]
> $$
>
> where $\lambda = 10$ is used to balance the scale between MSE loss and contrastive loss, $e^t$ and $e^a$ is the text and audio embeddings, respectively, $N = 384$ is the batch size, $s_{i,j}$ is the similarity score calculated as the dot product of normalized text embedding $e^t_i$ and audio embedding $e^a_j$, and $\tau = 0.05$ is the temperature. This loss aligns both the scale and direction of the audio and text embeddings.

---

> > ### Author Response · Authors · 2023-11-22
> > **Author Response to Reviewer oFtB (2/5) Audio-Text Alignment**
> >
> > ---
> > **Question 1. Audio-Text Alignment** (cont.)
> >
> > | Model                                          |   ESC50  |   DCASE  |    VS    |    TUT   |    BJO   |    VGG   |    FSD   | AudioSet | Classif. Avg. | AudioCaps |  Clotho  | Cap. Avg. | Audio Question Instruction Following Rate | Language Question Instruction Following Rate |
> > |------------------------------------------------|:--------:|:--------:|:--------:|:--------:|:--------:|:--------:|:--------:|:--------:|:-------------:|:---------:|:--------:|:---------:|:----------:|:-------------:|
> > | Linear Layer w/ Alignment Training             | **83.2** |   45.3   | **59.5** |   30.4   |   64.4   |   50.0   | **46.5** |   18.3   |      49.7     |    16.8   | **11.9** |    14.4   |    86.5    |      93.2     |
> > | Transformer Layer w/  Alignment Training       |   80.6   | **46.8** |   55.8   |   30.5   |   61.9   | **51.9** |   45.7   |   16.7   |      48.7     |    16.8   |   11.5   |    14.2   |  **89.9**  |      93.1     |
> > | Linear Layer wo/  Alignment Training (Default) |   83.1   |   45.9   |   55.6   | **32.5** | **69.9** |   50.3   |   46.3   | **18.7** |    **50.3**   |   **17.0**  | **11.9** |  **14.5** |    87.6    |    **96.9**   |
> >
> > We present the results in the above Table and observe that for the same linear projection layer, additional audio-text alignment training does not significantly impact model performance. Moreover, scaling up the alignment module to a Transformer layer results in even poorer performance. We attribute this to several reasons:
> >
> > - In the original LTU training, the model is trained with 5.6 million samples over multiple epochs, and the audio encoder is set as trainable for three stages of the training. Consequently, during the original LTU training process, the audio encoder is sufficiently trained to bridge the gap with the large language model, additional alignment training unnecessary.
> >
> > - Increasing the capacity of the alignment module would introduce a larger number of randomly initialized parameters, complicating the training process. Considering that the AST audio encoder is a 12-layer Transformer and was set as trainable in the original LTU training, an increase in the capacity of the alignment module is not necessary.
> >
> > - The audio embeddings serve as soft prompts for large language models and do not necessarily need to be in the text embedding space. Therefore, additional audio-text training does not provide a more effective initialization for LTU training.
> >
> > We have added these experiments to **Section F Discussion on the Audio and Text Embedding Space Alignment** in the revised paper. We thank the reviewer again for this very valuable comment and hope that the extended discussion will offer additional insights to our readers.

---

> > > ### Author Response · Authors · 2023-11-22
> > > **Author Response to Reviewer oFtB (3/5) Compare LTU with Pengi**
> > >
> > > **Question 2. Compare with Pengi**
> > >
> > > ------
> > > >The concurrent works show higher performances. Compared to Pengi, the closed-ended audio task performances are lower. I understand that the proposed paper is focusing on the open-ended problem, but it would be better to elaborate more in detail that the proposed paper is competitive compared to the concurrent works.
> > >
> > > We thank the reviewer for the comment and agree that comparing LTU with concurrent work would better place LTU in the context. The reviewer is correct that Pengi is better on some closed-ended benchmarks (e.g., ESC-50), but as we will show below, on average, LTU is **slightly better** than Pengi on closed-ended tasks and significantly better for open-ended tasks.
> > >
> > > We compare LTU with Pengi in three aspects.
> > >
> > > ------
> > > ### 1. Technical Similarity and Difference
> > >
> > > **Similarity**:
> > > - Architecture-wise, both Pengi and LTU connects an audio encoder with an autoregressive language model. Both models can generate text from the given audio and text prompt.
> > > - Performance-wise, both Pengi and LTU can do zero-shot predictions on unseen datasets. On closed-ended tasks, Pengi and LTU perform similarly.
> > >
> > > **Difference**
> > > - **Motivation:** Pengi focuses on *transfer learning*, i.e., using a single model for 8 audio perception tasks, while LTU focuses on *unifying audio perception with understanding*. Unlike LTU, Pengi is not designed for audio understanding and answering *free-form open-ended* questions from users.
> > > - **Language Model Scale:** Pengi employs GPT2-Base (124M parameters) as its language model, whereas LTU uses LLaMA (7B parameters), which is over 50x larger than GPT2-Base. Additionally, LLaMA is trained with significantly more data than GPT2. Scaling up the language model substantially enhances LTU's understanding capabilities.
> > > - **Training Data:** Pengi is trained on multiple audio datasets, and directly uses the original text-form labels as answers for AQA. This approach is similar to the closed-ended part of our OpenAQA dataset. However, OpenAQA also includes another 3.7M open-ended AQAs generated with the proposed *audio instruction generation* method (not generated from a template). The open-ended portion of our training data is pivotal in enabling LTU to answer *any free-form open-ended* questions.
> > > - **Training Curriculum:** To better unifying audio perception and generation, LTU adopts a unique perception-to-understanding training curriculum.
> > > - **Open-Ended Evaluation:** LTU undergoes evaluation for both closed-ended and open-ended tasks, incorporating both subjective and objective evaluations. Conversely, Pengi is evaluated solely with closed-ended tasks. Please note that the definition of an open-ended task differs between the Pengi paper and this paper; Pengi considers audio captioning as an open-ended task, while we categorize them as closed-ended tasks.
> > >
> > > ---
> > > ### 2.Closed-Ended Task Performance Comparison
> > >
> > > |               |   ESC50  |  FSD50K  |   DCASE  |    TUT   | Beijing Opera | Vocal Sound |  Average |
> > > |---------------|:--------:|:--------:|:--------:|:--------:|:-------------:|:-----------:|:--------:|
> > > | Pengi         | **91.9** | **46.7** |   33.8   | **35.2** |      62.3     |     60.3    |   55.1   |
> > > | LTU (Default) |   83.1   |   46.3   |   45.9   |   32.5   |    **69.9**   |     55.6    |   55.6   |
> > > | LTU (Full FT) |   85.9   |   45.7   | **47.2** |   33.0   |      59.8     |   **69.0**  | **56.7** |
> > >
> > > We summarize the performance of LTU and Pengi on closed-ended audio classification tasks in the above Table (using the result reported by the Pengi paper). Pengi and LTU each win on three out of the six tasks. In general, LTU with the default LoRA setting performs similarly to Pengi (average score 55.6 vs 55.1), while the fully finetuned version of LTU performs slightly better, achieving an average score of 56.7.

---

> > > > ### Author Response · Authors · 2023-11-22
> > > > **Author Response to Reviewer oFtB (4/5) Compare LTU with Pengi**
> > > >
> > > > **Question 2. Compare with Pengi** (cont.)
> > > >
> > > > ---
> > > > ### 3. Open-Ended Task Performance Comparison
> > > >
> > > > |               | GPT-3.5 Generated Questions | GPT-4 Generated Questions | Human Generated Questions |
> > > > |---------------|:---------------------------:|:-------------------------:|:-------------------------:|
> > > > | Pengi         |             7.4             |            3.1            |            28.3           |
> > > > | LTU (Default) |           **95.9**          |          **96.9**         |          **70.1**         |
> > > >
> > > > We use GPT-4 (with the prompt ``Below is a pair of question and response. Identify if the response answers the question. Return yes or no.``) to evaluate the instruction following rate of LTU and Pengi in response to open-ended questions and present the results in the above Table. With a stronger language model and training with the OpenAQA dataset, LTU dramatically outperforms Pengi.
> > > >
> > > > However, please note that Pengi is neither designed nor trained for open-ended question answering and understanding, thus this task is not entirely fair to Pengi. We conduct this experiment just to highlight the differences between LTU and Pengi.
> > > >
> > > > ---
> > > > ### Summary
> > > >
> > > > To summarize, while LTU and Pengi share some similarities in architecture, they have different motivations and designs. They perform similarly in closed-ended audio classification tasks, but LTU is more capable of answering open-ended questions and demonstrating audio understanding.
> > > >
> > > > We have added this discussion to **Section G Compare with Pengi** in the revised paper. We thank the reviewer again for this comment and hope the discussion will help the reader understand the difference between LTU and concurrent works.

---

> > > > > ### Author Response · Authors · 2023-11-22
> > > > > **Author Response to Reviewer oFtB (5/5) Minors**
> > > > >
> > > > > **Question 3. Dimension of the audio input**
> > > > >
> > > > > ------
> > > > > >I don't understand 64(time) X 16(frequency) in the 5th line of Audio Encoder part in Section 2: LTU Model Architecture. Is it typo(I think it should be 8)? Or what is the meaning of frequency 16?
> > > > >
> > > > > The reviewer is totally correct on this. It is a typo and should be 8, we have fixed this in the revised paper.
> > > > >
> > > > > ------
> > > > > >I want to know how the dataset is formulated (Q and A) that is generated from GPT.
> > > > >
> > > > > We thank the reviewer for the comment. In Table 2 of the paper, we show the prompt for GPT-3.5-Turbo, but due to the space limitation, we omit the formatting instruction ``Format each QA pair in a single line as a JSON dictionary (key "q'' for question, and "a'' for answer, wrapped with { and }). Do not include any other explanation.``, which may cause confusion, we apologize for this.
> > > > >
> > > > > The full prompt is
> > > > >
> > > > > >Based on the following audio clip, generate 10 different types of complex open-ended questions that require step-by-step thinking, and corresponding step-by-step answers.
> > > > > The following information is provided: the sound events appear in the audio clip, together with its acoustic features, and corresponding onset and offset time stamps. A description of the content of the audio clip is also provided.
> > > > > Questions should be about the audio, e.g., which sound event is recognized and why (e.g., based on its acoustic feature), what can be inferred based on the combination of sound events; the temporal relationship between the sound events and what can be inferred from that; the potential scenario that such an audio clip could happen, if the audio clip is special (e.g., urgent, funny, interesting, abnormal, unique, etc) and why, what mood or atmosphere this audio clip conveys, etc. **Format each QA pair in a single line as a JSON dictionary (key "q'' for question, and "a'' for answer, wrapped with { and }). Do not include any other explanation.**
> > > > >
> > > > > The output of GPT is a JSON dictionary looks like
> > > > > ``
> > > > > [{
> > > > >   "q": "How would you describe the tone of the sound of the
> > > > >         accelerating engine?",
> > > > >   "a": "The tone of the sound of the accelerating engine is
> > > > >         high-pitched, short and intense."
> > > > > },
> > > > > ...
> > > > > {
> > > > >   "q": "What is the acoustic feature that distinguishes the
> > > > >         sound of the ambulance siren from the generic impact sounds?",
> > > > >   "a": "The acoustic feature that distinguishes the sound of the
> > > > >         ambulance siren from the sound of generic impact sounds is that
> > > > >         the former is high-pitched and wailing, while the latter is
> > > > >         loud and sharp."
> > > > > }]
> > > > > ``
> > > > > We then extract the questions and answers from the JSON dictionary. In the revised paper, we added a new section **Appendix M Full GPT-3.5-Turbo Prompt and Sample Outputs** to show the full prompt and sample JSON output. We also added a note in Table 2 to indicate the prompt is an abbreviated version due to the space limitation. We hope these changes will make the description of our data generation process more clear.
> > > > >
> > > > > ---
> > > > >
> > > > > **Other Improvements in the Revision:** In addition to the above-mentioned changes, we have incorporated new experiments in response to the comments from other reviewers. These are detailed in Appendix E, which discusses scaling up the LLM, and Appendix H, which focuses on evaluating LTU in temporal analysis tasks. We would greatly appreciate it if the reviewer could take the time to review these new sections.
> > > > >
> > > > > ---
> > > > > We hope we addressed all your concerns, and in this case would be happy if the reviewer could kindly consider increasing the score, and if not we are more than willing and happy to engage in a discussion with the reviewer to answer further questions and/or provide more clarifications as needed.

---

> > > > > > ### Comment · Reviewer_oFtB · 2023-11-22
> > > > > > **Thank you for the author response**
> > > > > >
> > > > > > I appreciate the authors' comprehensive explanation of my questionnaires and the additional experiments regarding my concern. Based on the thorough additional experimental results and clear explanation, I would raise my rating. Please update the paper based on the rebuttal.

---

> > > > > > > ### Author Response · Authors · 2023-11-22
> > > > > > >
> > > > > > > Dear Reviewer oFtB,
> > > > > > >
> > > > > > > Thank you very much for your prompt and encouraging response (and increasing the score)!
> > > > > > >
> > > > > > > We also appreciate your reminder to update the revised paper. We have just done that and you should be able to see it by clicking the top-right PDF button.
> > > > > > >
> > > > > > > To make it easier to track the new sections, we temporarily put all of them in Appendix but will consider moving them to the main manuscript in the next version.
> > > > > > >
> > > > > > > Thanks again.

---

### Official Review · Reviewer_2qgj · 2023-10-31

**Soundness:** 4 excellent
**Presentation:** 4 excellent
**Contribution:** 4 excellent
**Rating:** 8
**Confidence:** 5

**Summary:**

The authors present an Audio Language Model (ALM) capable of understanding generic audio signals. The system in its current form is not designed for speech or music transcription but to provide a generic description/caption summarizing the audio. The crux of the problem to solve in such models has to do with text generation conditioned on audio signal. The fundamental strategy of approaches in this realm has been to interface an audio encoder somehow with a Large Language Model (LLM) for audio conditioned text generation.

In this work the authors propose to use Audio Spectrogram Transformer (AST) as the Audio Encoder and interface it with the LLaMA model using a simple projection layer in between with LoRA (Low RAnk) adapters added on top of LLaMA model. As has been the common practice such a network is trained on multiple audio related tasks by unifying the input output representation of the network in the form of [Audio, Question] as input to [Answer] as output format. To train this model the authors curate data from popular public datasets across 8 different tasks and augment the same audio with multiple Question answer pairs. The Question-Answer pairs are generated in 2 buckets - Close-Ended and Open Ended. In the Close-Ended bucket Questions are paraphrased using GPT3.5-Turbo for diversity. Answers are generated using a rule based system. In the Open Ended bucket Questions and Answers are generated using GPT-3.5 based on Audi metadata (audio events, captions, acoustic features, temporal information) and prompt engineering. The LTU model is then trained solely on the OpenAQA Dataset with an interesting Curriculum training. the gist of curriculum training is to go from simple to complex. Train the projection layer ---> Train AST + Projection layer + LoRA with frozen LLaMa. In terms of tasks: Simpl close ended tasks  --> Close + Open ended tasks.

Overall the paper is very well written. The authors have done an excellent job in presenting the idea, exploring the idea with effective ablation studies, a straight-forward depiction of results with comparison to the right baselines and providing a plausible explanation to interpret the results in the context of baselines.

**Strengths:**

Originality:
1. The idea, at the time the paper was originally written, was indeed very novel as there were not many audio language models around back in May. The idea makes a lot of sense.
2. Audio Instruction Generation (AIG) is also quite nice and interesting.
3. The curriculum learning is yet another contribution which makes a lot of sense and the authors have proposed an intuitive curriculum and backed it up with apt ablation study to show its utility.
4. The Human evaluation study adds a ton of value in judging LTU in my perspective.


Quality:
1. A high quality manuscript with sufficient experimental details.
2. I thoroughly enjoyed reading the full paper including all the Appendix sections. Very thoughtfully done experimental ablation study.

Clarity:
1. Very well written manuscript with clear non-monotonic description of details.

Significance:
1. In my opinion this is a significant paper as it explores one of the straight-forward ways to couple an audio encoder with a trained LLM and carefully examines this coupling from several different viewpoints and contributes the OpenAQA dataset which can be a useful public resource for future research. The curriculum training is yet another aspect that makes this effort worthy as it shows that a brute force approach to just wrap in all possible audio-text paired data may not be as good overall.

**Weaknesses:**

1. It is not clear to me what the value of OpenAQA dataset is on top of of the textual metadata available with most of these datasets. It would have been fascinating if the authors were to do an ablation study to train their model in the format of (Audio, Text) --> Text format - something similar to what PENGI does. Use the text data that comes with each dataset as is and compare this with :
  a) Using OpenAQA (current setting)
  b) Augmenting the original audio text pairs with OpenAQA

2. It might be valuable to evaluate PENGI on Open ended tasks. My hunch is LTU would be far better than Pengi in open ended tasks although Pengi might be better on Close-ended tasks. This would probably highlight LTU significantly as the two approaches are contemporary in many ways and are very similar in the overarching goal. However the approaches taken are different. I realize that PENGI's checkpoint was probably not available when this paper was submitted but it is now. I would encourage the authors to do this comparison to show how LTU could be a more generic model which can be super useful for users to interact with an audio language model through natural text.

**Questions:**

Question raised in Weakness section.

---

> ### Author Response · Authors · 2023-11-22
> **Author Response to Reviewer 2qgj (1/3) Value of OpenAQA**
>
> Dear Reviewer 2qgj,
>
> We sincerely thank you for your very positive feedback and are gratified to recognize from your insightful comments a deep comprehension of our paper's content and objectives. Let us respond to your questions point by point.
>
> ---
> **Question 1. Value of OpenAQA**
>
> >It is not clear to me what the value of OpenAQA dataset is on top of of the textual metadata available with most of these datasets. It would have been fascinating if the authors were to do an ablation study to train their model in the format of (Audio, Text) --> Text format - something similar to what PENGI does. Use the text data that comes with each dataset as is and compare this with : a) Using OpenAQA (current setting) b) Augmenting the original audio text pairs with OpenAQA.
>
> We thank the reviewer for the question. As the reviewer pointed out in the summary, the OpenAQA consists of two parts: the closed-ended part (the question is paraphased by GPT but the answer is generated with a rule based algorithm based on ground truth labels), and the open-ended part.
>
> The closed-ended part of OpenAQA is actually very similar to the Pengi training data:
> - Both the OpenAQA closed-ended dataset and the Pengi dataset use a template to formulate answers. For instance, in Pengi, the format for audio event classification is ``{event a}, {event b},...`` (e.g., ``Ambulance, Traffic noise, Accelerating``), while in OpenAQA, it is ``Label: {event a}, {event b}`` (e.g., ``Label: Ambulance, Traffic noise, Accelerating``).
> - The differences between the OpenAQA closed-ended dataset and the Pengi dataset are twofold:
>   - Pengi has only **one** question (input prompt) for each task, such as ``this is a sound of`` for audio event classification. In contrast, the OpenAQA closed-ended dataset paraphrases the question **multiple** times (about 100 times) using GPT-3.5-Turbo. Example questions for the audio event classification task include ``Generate audio labels``, ``Produce audible tags``, and ``Generate audio annotations``. This approach enhances LTU's robustness to question perturbations. The Pengi paper (Appendix C and Table 17) reports varying model performance with different input prompts. Our design aims to mitigate this issue.
>   - OpenAQA uses GPT-3.5-Turbo to generate acoustic feature descriptions, which are then utilized to train the LTU model. This enhances its ability to recognize acoustic features alongside sound classes, allowing LTU to generalize predictions for unseen sound classes based on their acoustic features. An example is ``high-pitched, rapidly changing frequency`` for the ``ambulance`` class, leading to an answer like ``high-pitched, rapidly changing frequency → Ambulance``. Pengi does not have this design.
>
> However, these differences are minor. Essentially, the closed-ended part of OpenAQA and the Pengi training data are generated using very similar methods.
>
> The major difference between LTU and Pengi is the **open-ended part of OpenAQA**. Here, both the question and answer are generated by **GPT**, as opposed to a rule-based algorithm or template. We present examples in Table 2 of the paper:
>
> ``Question: What can be inferred from the fact that traffic noise fades while the ambulance siren echoes?``
>
> ``Answer: It can be inferred that the ambulance is approaching or near the location where the audio clip was recorded, and that the traffic is yielding to the emergency vehicle.``
>
> ``Question: What mood or atmosphere does the audio clip convey?``
>
> ``Answer: The audio clip conveys a sense of urgency, due to the sound of the ambulance siren.``
>
> The answers to open-ended questions are not taken directly from the text-form labels of the original dataset, but are instead derived by GPT through background knowledge and reasoning, based on the ground truth labels. As shown in Figure 3 of the paper, over 95% of the questions in the 3.7M dataset are unique, covering a large portion of the question space. This enables LTU to answer **any** question, not just ones it has previously seen. In contrast, when Pengi encounters a new question, it struggles to respond appropriately, for example:
>
> ``Question: What can be inferred from the combination of mechanical, male speech, and light engine sound events?``
>
> ``Pengi: two``
>
> ``LTU: It is likely that there are multiple machines or vehicles in operation nearby.``
>
> ---
> In summary, while the closed-ended part of OpenAQA is very similar to Pengi's training data, with only minor differences, the key distinction between OpenAQA and Pengi lies in OpenAQA's open-ended section, where answers are not based on original labels. It is this open-ended aspect that enables LTU to effectively address open-ended questions.

---

> ### Author Response · Authors · 2023-11-22
> **Author Response to Reviewer 2qgj (2/3) Value of OpenAQA**
>
> **Question 1. Value of OpenAQA** (cont.)
>
> In our paper, we conduct experiments to compare models trained with **closed-ended data only** (akin to Pengi) with those trained with **both closed-ended and open-ended data** (default LTU). We show results in Table 5 right and Table 14 of the paper. These can be summarized as the following table.
>
> |                  Training Setting                  | Closed-Ended Audio Classification Performance | Open-ended Question Instruction Following Rate |
> |:--------------------------------------------------:|:---------------------------------------------:|:----------------------------------------------:|
> | Closed-ended data only (stage 1,2,3)               |                      37.1                      |                      22.5                      |
> | Closed-ended data only (same training iterations)  |                      47.3                      |                        -                       |
> | Closed-ended and open-ended data (default)         |                      50.3                      |                      96.9                      |
>
> These results demonstrate that including the open-ended section of OpenAQA leads to improvements in performance for both closed-ended and open-ended tasks. Specifically:
> - For closed-ended tasks, the inclusion of open-ended training results in enhanced performance (from 47.3 to 50.3), indicating that *learning to understand can bolster perception ability.*
> - For open-ended tasks, the model trained only with closed-ended QAs shows a markedly lower instruction following rate compared to the one trained with both types of data (22.5% vs 96.9).
>
> This evidence justifies the importance of utilizing GPT to generate open-ended QAs in our training regimen. We hope our explanation and experiments could address the reviewer's question.
>
> ---
> **Question 2. Compare with Pengi**
>
> We thank the reviewer for the comment and for pointing us to the pretrained Pengi checkpoint. We were able to download the checkpoint and run experiments based on it. We agree that comparing LTU with concurrent work would better place LTU in the context. The reviewer is correct that Pengi is better on some closed-ended benchmarks (e.g., ESC-50), but as we will show below, on average, LTU is **slightly better** than Pengi on closed-ended tasks and significantly better for open-ended tasks.
>
> We compare LTU with Pengi in three aspects.
>
> ------
> ### 1. Technical Similarity and Difference
>
> **Similarity**:
> - Architecture-wise, both Pengi and LTU connects an audio encoder with an autoregressive language model. Both models can generate text from the given audio and text prompt.
> - Performance-wise, both Pengi and LTU can do zero-shot predictions on unseen datasets. On closed-ended tasks, Pengi and LTU perform similarly.
>
> **Difference**
> - **Motivation:** Pengi focuses on *transfer learning*, i.e., using a single model for 8 audio perception tasks, while LTU focuses on *unifying audio perception with understanding*. Unlike LTU, Pengi is not designed for audio understanding and answering *free-form open-ended* questions from users.
> - **Language Model Scale:** Pengi employs GPT2-Base (124M parameters) as its language model, whereas LTU uses LLaMA (7B parameters), which is over 50x larger than GPT2-Base. Additionally, LLaMA is trained with significantly more data than GPT2. Scaling up the language model substantially enhances LTU's understanding capabilities.
> - **Training Data:** Pengi is trained on multiple audio datasets, and directly uses the original text-form labels as answers for AQA. This approach is similar to the closed-ended part of our OpenAQA dataset. However, OpenAQA also includes another 3.7M open-ended AQAs generated with the proposed *audio instruction generation* method (not generated from a template). The open-ended portion of our training data is pivotal in enabling LTU to answer *any free-form open-ended* questions.
> - **Training Curriculum:** To better unifying audio perception and generation, LTU adopts a unique perception-to-understanding training curriculum.
> - **Open-Ended Evaluation:** LTU undergoes evaluation for both closed-ended and open-ended tasks, incorporating both subjective and objective evaluations. Conversely, Pengi is evaluated solely with closed-ended tasks. Please note that the definition of an open-ended task differs between the Pengi paper and this paper; Pengi considers audio captioning as an open-ended task, while we categorize them as closed-ended tasks.

---

> > ### Author Response · Authors · 2023-11-22
> > **Author Response to Reviewer 2qgj (3/3) Compare with Pengi**
> >
> > **Question 2. Compare with Pengi** (cont.)
> >
> > ---
> > ### 2. Closed-Ended Task Performance Comparison
> >
> > |               |   ESC50  |  FSD50K  |   DCASE  |    TUT   | Beijing Opera | Vocal Sound |  Average |
> > |---------------|:--------:|:--------:|:--------:|:--------:|:-------------:|:-----------:|:--------:|
> > | Pengi         | **91.9** | **46.7** |   33.8   | **35.2** |      62.3     |     60.3    |   55.1   |
> > | LTU (Default) |   83.1   |   46.3   |   45.9   |   32.5   |    **69.9**   |     55.6    |   55.6   |
> > | LTU (Full FT) |   85.9   |   45.7   | **47.2** |   33.0   |      59.8     |   **69.0**  | **56.7** |
> >
> > We summarize the performance of LTU and Pengi on closed-ended audio classification tasks in the above Table (using the result reported by the Pengi paper). Pengi and LTU each win on three out of the six tasks. In general, LTU with the default LoRA setting performs similarly to Pengi (average score 55.6 vs 55.1), while the fully finetuned version of LTU performs slightly better, achieving an average score of 56.7.
> >
> > ---
> > ### 3. Open-Ended Task Performance Comparison
> >
> > |               | GPT-3.5 Generated Questions | GPT-4 Generated Questions | Human Generated Questions |
> > |---------------|:---------------------------:|:-------------------------:|:-------------------------:|
> > | Pengi         |             7.4             |            3.1            |            28.3           |
> > | LTU (Default) |           **95.9**          |          **96.9**         |          **70.1**         |
> >
> > We use GPT-4 (with the prompt ``Below is a pair of question and response. Identify if the response answers the question. Return yes or no.``) to evaluate the instruction following rate of LTU and Pengi in response to open-ended questions and present the results in the above Table. With a stronger language model and training with the OpenAQA dataset, LTU dramatically outperforms Pengi. This also answers the reviewer's first question regarding the value of OpenAQA.
> >
> > However, please note that Pengi is neither designed nor trained for open-ended question answering and understanding, thus this task is not entirely fair to Pengi. We conduct this experiment just to highlight the differences between LTU and Pengi.
> >
> > ---
> > ### Summary
> >
> > To summarize, while LTU and Pengi share some similarities in architecture, they have different motivations and designs. They perform similarly in closed-ended audio classification tasks, but LTU is more capable of answering open-ended questions and demonstrating audio understanding.
> >
> > We have added this discussion to **Section G Compare with Pengi** in the revised paper. We thank the reviewer again for this comment and hope the discussion will help the reader understand the difference between LTU and concurrent works.
> >
> > ---
> > **Other Improvements in the Revision:** In addition to the above-mentioned changes, we have incorporated new experiments in response to the comments from other reviewers. These are detailed in Appendix E, which discusses scaling up the LLM, Appendix F, which discusses audio-text pretraining, and Appendix H, which focuses on evaluating LTU in temporal analysis tasks. We would greatly appreciate it if the reviewer could take the time to review these new sections.
> >
> > ---
> > We hope we addressed all your concerns, and if not we are more than willing and happy to engage in a discussion with the reviewer to answer further questions and/or provide more clarifications as needed.

---

### Official Review · Reviewer_1UHZ · 2023-11-07

**Soundness:** 3 good
**Presentation:** 3 good
**Contribution:** 3 good
**Rating:** 6
**Confidence:** 5

**Summary:**

This paper proposes a multimodal large language model with the ability of general audio perception. The model combines the AST audio encoding frontend with the LLaMA large language model, utilizing LoRA for fine-tuning, resulting in a model with both audio perception and reasoning capabilities. Moreover, this paper constructed a large-scale dataset for this task, encompassing closed-ended and open-ended Q&A pairs. Extensive experiments illustrate that this model exhibits outstanding performance across various audio-related tasks.

**Strengths:**

1.The paper introduces for the first time a large language model that combines both general audio perception capabilities and language reasoning abilities, along with the datasets used for training. It is highly innovative and holds significant importance for the development of general artificial intelligence.

2.Through a significant number of ablation experiments, this paper extensively researched the model's hyperparameter configurations and training strategies, offering highly instructive guidance for related work.

3.The model excels in various audio-related tasks and open-ended question answering, demonstrating its outstanding performance.

**Weaknesses:**

1.The performance of this model is closely related to both the AST encoding frontend and the LLaMA model's performance. Ablation studies on the varying parameter counts of these two components would be valuable, if possible.

2.The authors may add subjective evaluations to the ablation experiments to better demonstrate that the LoRA fine-tuning strategy mitigates catastrophic forgetting issues.

**Questions:**

1.Why is the results of LTU on open-ended problems preferred compared to those of GPT-4 in Section 5.2.2? In terms of the data generation method of open-ended Q-A pairs, LTU's inferential knowledge appears to be transferred from the data generation model, i.e. GPT 3.5 Turbo model. These experimental results seem to indicate that the performance of the student model is superior to that of the teacher model (family).

2.Did the authors investigate whether errors (or hallucinations) in open-ended question generation model (i.e. GPT 3.5 turbo) could impact the performance of the LTU? In other words, is the correctness of the Q&A pairs generated by GPT from audio metadata reliable enough?

---

> ### Author Response · Authors · 2023-11-22
> **Author Response to Reviewer 1UHZ (1/3) Impact of the model size**
>
> Dear Reviewer 1UHZ,
>
> Thank you for the positive feedback and all these very valuable and constructive questions/suggestions! Let us respond to your questions point by point.
>
> ---
> **Question 1 - Impact of model size**
>
> >The performance of this model is closely related to both the AST encoding frontend and the LLaMA model's performance. Ablation studies on the varying parameter counts of these two components would be valuable, if possible.
>
> We thank the reviewer for this very constructive suggestion. We conduct the following new experiments to study the impact of model size.
>
> ---
> **1. Impact of the size of the large language model**
>
> Throughout the original paper, we use LLaMA model consisting of 7 billion parameters (LLaMA-7B) as the LTU language model. In this revision, we conduct experiments on LTU based on LLaMA with 13 billion parameters (LLaMA-13B).
>
> | Size |   ESC50  |   DCASE  |    VS    |    TUT   |    BJO   |    VGG   |    FSD   | AudioSet | Classif. Avg. | AudioCaps |  Clotho  | Cap. Avg. | Audio Question Instruction Following Rate | Language Question Instruction Following Rate |
> |------------|:--------:|:--------:|:--------:|:--------:|:--------:|:--------:|:--------:|:--------:|:-------------:|:---------:|:--------:|:---------:|:-----------------------------------------:|:--------------------------------------------:|
> | LLaMA-7B   |   83.1   | **45.9** |   55.6   | **32.5** | **69.9** |   50.3   |   46.3   | **18.7** |    **50.3**   |  **17.0** |   11.9   |    14.5   |                    **96.9**                   |                     87.6                     |
> | LLaMA-13B  | **83.6** |   44.7   | **64.2** |   31.0   |   52.1   | **50.7** | **46.8** |   17.9   |      48.9     |    16.9   | **12.1** |  **14.5** |                    95.4  |     **89.6**    |
>
> As indicated in the table above, we observe a slight decrease in audio classification performance with the increasing size of the LLaMA model while the captioning performance improves slightly with larger model sizes. There is no significant difference in the instruction following rate for pure language and audio open-ended questions. However, we note that the 13B model tends to provide more rational answers due to its more extensive knowledge base, as exemplified in the following sample:
>
> ``Question: How would the audio clip be different if the power tool was being used in a larger or open space?``
>
> ``LTU-7B: The sound of the power tool would likely be louder and more noticeable in an open space.`` (wrong)
>
> ``LTU-13B: If the power tool was being used in a larger or open space, the sound of the power tool would likely be more diffuse and less intense than it is in this clip.`` (correct)
>
> ---
> **2. Impact of the size of the audio projection module**
>
> Since large language model training is expensive, we can only train a limited number of models with our computational resources. Thus, we started with the strongest audio encoder we could find when we prepared this work (AST with CAV-MAE pretraining and finetuning). A smaller audio encoder could lead to similar performance, but unlikely to outperform the larger audio encoder. In addition, the number of parameters of our audio encoder is just about 1.3% of the LLaMA language model.
>
> The original LTU utilizes a linear layer to project audio embeddings to the large language model input, which might be overly simple. Therefore, in this revision, we consider to use a Transformer layer instead. To ensure a fair comparison, both the linear and Transformer layers have been pretrained using audio-text contrastive learning, as detailed in Appendix F.
>
> |  Model                                        |   ESC50  |   DCASE  |    VS    |    TUT   |    BJO   |    VGG   |    FSD   | AudioSet | Classif. Avg. | AudioCaps |  Clotho  | Cap. Avg. |  Audio Question Instruction Following Rate | Language Question Instruction Following Rate |
> |------------------------------------------------|:--------:|:--------:|:--------:|:--------:|:--------:|:--------:|:--------:|:--------:|:-------------:|:---------:|:--------:|:---------:|:----------:|:-------------:|
> | Linear Layer              | **83.2** |   45.3   | **59.5** |   30.4   |   **64.4**   |   50.0   | **46.5** |   **18.3**   |  **49.7**   |   16.8   | **11.9** |    **14.4**   |    86.5    |      **93.2**     |
> | Transformer Layer       |   80.6   | **46.8** |   55.8   |   **30.5**   |   61.9   | **51.9** |   45.7   |   16.7   |    48.7   |    16.8   |   11.5   |    14.2   |  **89.9**  |      93.1     |
>
> As shown in the table, expanding the layer capacity results in decreased performance, which can be attributed to the introduction of a larger number of randomly initialized parameters that complicates the training process. We thus conclude that the original configuration of a linear layer is a good design choice.
>
> We thank the reviewer again for the insightful comment, we have added these discussions to Appendix E and F of the paper.

---

> > ### Author Response · Authors · 2023-11-22
> > **Author Response to Reviewer 1UHZ (2/3) Why LTU outperforms GPT-4?**
> >
> > **Question 2 - Evaluating the catastrophic forgetting difference?**
> > >The authors may add subjective evaluations to the ablation experiments to better demonstrate that the LoRA fine-tuning strategy mitigates catastrophic forgetting issues.
> >
> > We thank the reviewer for the question.
> >
> > In Table 5 of the paper, we show that full finetuning would cause a dramatic instruction following rate drop on pure language tasks.
> >
> > | Training Setting  | Average Audio Classification Performance | Pure Language Task Instruction Following Rate |
> > |-------------------|:----------------------------------------:|:---------------------------------------------:|
> > | LoRA (lr=1e-4)    |                   50.3                   |                      87.6                     |
> > | Full FT (lr=1e-4) |                   50.4                   |                      35.0                     |
> > | Full FT (lr=2e-5) |                   51.3                   |                      53.5                     |
> >
> > The instruction following rate is evaluated by GPT-4 with the prompt ``Below is a pair of question and response. Identify if the response answers the question. Return yes or no.`` While GPT-4 may not be a perfect tool for evaluation, a notable difference of approximately 35% clearly indicates a significant drop in performance. The following example illustrates a case where LTU-Full-FT fails.
> >
> > ``Question: Generate a shopping list for a fast food meal. `` (no audio is provided)
> >
> > ``LTU-LoRA: Here is the shopping list for a fast food meal:\n\n* A burger\n* French fries\n* A drink (such as soda or water)\n* Condiments (such as ketchup, mustard, and mayo)\n* Napkins\n* Straws (if desired)\n\nThis list can be adjusted based on personal preferences and dietary restrictions.``
> >
> > ``LTU-Full-FT: The audio clip does not provide any information about the specific items on the shopping list.``
> >
> > For open-ended audio tasks, we also observe that the fully-finetuned LTU tends to forget its original knowledge, resulting in shorter and less diverse responses. For audio questions generated by GPT-4, the fully-finetuned LTU's average answer word length is 26 and its vocabulary size is 8,456, whereas the LoRA version shows an average word length of 35 and a vocabulary size of 9,621.
> >
> > ---
> > **Question 3 - Why LTU outperforms GPT-4?**
> >
> > >Why is the results of LTU on open-ended problems preferred compared to those of GPT-4 in Section 5.2.2? In terms of the data generation method of open-ended Q-A pairs, LTU's inferential knowledge appears to be transferred from the data generation model, i.e. GPT 3.5 Turbo model. These experimental results seem to indicate that the performance of the student model is superior to that of the teacher model (family).
> >
> > We thank the reviewer for the question. The reason why LTU can outperform GPT-4 is because the input to LTU is **raw audio** while the input to GPT-4 is **text-form audio event label, timestamps, and caption**. Raw audio inherently contains richer information compared to its text-form counterpart. The limitation of GPT-4 not being able to process audio directly (which is a key motivation for this work) restricts its use to text-based inputs only.
> >
> > We summarize the model input and language model for LTU and GPT-4 in the following tables:
> >
> > |                    |                                                    |                             |
> > |:------------------:|:--------------------------------------------------:|:---------------------------:|
> > | **LTU Evaluation** |                **Input: Raw Audio**                |      **LLM: LLaMA-7B**      |
> > | Advantage          |             Contain richer information             | Trained specially for audio |
> > | Disadvantage       | No ground truth is provided, LTU may make mistake. |      Smaller and weaker     |
> >
> > |                    |                                                    |                             |
> > |:------------------:|:--------------------------------------------------:|:---------------------------:|
> > | **GPT-4 Evaluation**    | **Input: Text-form Ground Truth Audio Lable, Time Stamps, and Caption** |          **LLM: GPT-4**          |
> > | Advantage    |               Ground truth information is provided.              |      Most advanced LLM      |
> > | Disadvantage |             Less information compared with raw audio             | Not trained for audio tasks |
> >
> > This comparison serves as an important purpose: one trivial solution for audio large language model is first to convert audio to text form, e.g., audio caption, and then feed it to a strong large language model. Our aim is to demonstrate that this method cannot surpass LTU's performance, primarily because audio captions inherently contain limited information. In the experiment, we allowed the baseline model to have an advantage by using the ground truth label and employing one of the strongest large language models, GPT-4. Despite this, the model still could not outperform LTU, thereby supporting our claim.

---

> > > ### Author Response · Authors · 2023-11-22
> > > **Author Response to Reviewer 1UHZ (3/3) Error caused by training data**
> > >
> > > ---
> > > **Question 4 - Errors caused by GPT generated data**
> > >
> > > >Did the authors investigate whether errors (or hallucinations) in open-ended question generation model (i.e. GPT 3.5 turbo) could impact the performance of the LTU? In other words, is the correctness of the Q&A pairs generated by GPT from audio metadata reliable enough?
> > >
> > > We appreciate the reviewer's insightful question, which is about the fundamental aspect of our core method audio instruction generation (AIG).
> > >
> > > As discussed in Section 3.2, AIG produces a broad spectrum of audio QAs, ranging from low-level tasks (like identifying audio events) to high-level understanding tasks (such as determining the atmosphere conveyed by the audio). And the errors can be broken down for low-level and high-level QAs.
> > >
> > > - For low-level QAs, we observe that GPT-3.5-Turbo almost never make an error, e.g., when the provided label is ``ambulance``, GPT-3.5-Turbo will not generate a QA with an answer like ``it is a police car''.
> > >
> > > - For high-level QAs, GPT-3.5-Turbo may make error/hallucination, e.g., when the provided label is ``ambulance``, GPT-3.5-Turbo typically associate it with an urgent situation while this may not be true for all audio clips containing ambulance.
> > >
> > > Therefore, we believe that for low-level questions, GPT-3.5-Turbo generated data does not introduce *additional* factual errors, as these QAs are based on ground truth labels. The factual errors in LTU's responses primarily stem from imperfections in its audio encoder. In contrast, the training data's hallucinations do impact LTU's performance on high-level questions.
> > >
> > > Given the challenge of automatically measuring errors in GPT-generated data due to its volume, we opted for a direct human evaluation of the LTU model. Table 7 in our paper presents the results of this evaluation, where human raters found 79.3% of LTU's answers to GPT-4 generated open-ended questions and 83.7% to human-generated open-ended questions completely correct. Overall, LTU's performance on open-ended questions is satisfactory.
> > >
> > > ---
> > >
> > > **Other Improvements in the Revision:** In addition to the above-mentioned changes, we have incorporated new experiments in response to the comments from other reviewers. These are detailed in Appendix F, which discusses audio-text alignment,  Appendix G, which focuses on comparing LTU with a concurrent audio language model Pengi, and Appendix H, which focuses on temporal analysis tasks. We would greatly appreciate it if the reviewer could take the time to review these new sections.
> > >
> > > ---
> > > We hope we addressed all your concerns, and in this case would be happy if the reviewer could kindly consider increasing the score, and if not we are more than willing and happy to engage in a discussion with the reviewer to answer further questions and/or provide more clarifications as needed.

---

### Author Response · Authors · 2023-11-21
**Author Response & Revision**

Dear Reviewers,

We appreciate your immensely helpful, insightful, and constructive comments on our paper
*Listen, Think, and Understand*. Following the reviewers' suggestions, we have supplemented a set of new experiments to address the reviewers' concerns. Please kindly check the **updated pdf file**. To make it easier to track the new sections, we temporarily put all of them in Appendix but will consider moving them to the main manuscript in the next version.

---
1. Following Reviewer 1UHZ's suggestions, we added experiments on scaling the LLaMA large language model from 7B to 13B. (**Appendix E - Scaling up the Large Language Model**)
2. Following Reviewer oFtB’s suggestions, we added experiments on additional audio-text alignment training before the LTU training. (**Appendix F - Discussion on the Audio and Text Embedding Space Alignment**)
3. Following Reviewer 2qgj and oFtB's suggestion, we added a detailed comparison with Pengi, a concurrent audio language model. (**Appendix G - Compare with Pengi**)
4. Following Reviewer wB1w’s suggestions, we added experiments on the temporal analysis capability of LTU. (**Appendix H - LTU Temporal Analysis Experiments**)
5. We added a list of GPT-generated acoustic feature description (**Table 17**) following reviewer wB1w's suggestion. We added the full GPT-3.5-Turbo prompt and sample output (**Section M - Full GPT-3.5-Turbo Prompt and Sample Outputs**) following reviewer oFtB's suggestion.
---
We hope the revised manuscript will better suit the ICLR conference, and are happy to discuss further with the reviewers and consider further revisions. We thank you for your continued interest in our research. In the following, under each reviewer's comment, we address the concerns of the reviewers point by point.

---

### Meta-Review · Area_Chair_veW7 · 2023-12-07

**Metareview:**

The authors present an LLM for audio perception and reasoning. 10 sec audio snippets are encoded using AST, and concatenated with text embeddings as input to the LLM. As LLM, the authors use the LLaMA model. LoRA adapters are used during the fine-tuning stage.

The main contribution of the work is the OpenAQA dataset, where various tasks are formulated in a question-answering format (audio, question, answer). Existing speech datasets including AudioSet, VGGSound, etc. are used as the training data, creating 845K unique audio clips. GPT3.5 is then used (along with available labels) to generate diverse questions, descriptions, etc. to create the final dataset. Another interesting aspect is the use of GPT3.5 to generate open ended questions and responses. Together, the dataset consists of ~5M training examples.

The reviewers commended task novelty, and the extensive evaluations / ablations that the authors perform.

The model is tied to the quality of AST, LLaMA and GPT3.5. Since these are pretraining, ablations around these are missing. Also, since the labels are generated using GPT, it is unclear how the errors in task generation affect overall quality. In their rebuttal, the authors addressed some of this: They conducted experiments using LLaMA 13B to show that there is not a significant difference in terms of qualitative evals, and using a more complex AST embedding transform.

Comparisons with Pengi came up, as a way of evaluating the necessity of generating the various datasets. The authors clarified that the open ended part of the dataset is what differentiates it from Pengi. They also compare to a version of the model that only uses close-ended data as a proxy for Pengi. They also directly compare Pengi, showing similar quality on close ended, but better quality on open ended tasks.

The paper went through a thorough author-reviewer discussion phase that improved the overall quality / score.

**Justification For Why Not Higher Score:**

The paper presents a simple and interesting way to combine audio encoder and LLMs, creating a large corpora for the task in the process. Given that the novelty is in the dataset creation and, the resulting model architectures are using prior models, a poster format is most appropriate.

**Justification For Why Not Lower Score:**

One of the first works that address open ended tasks using audio, and shows nice gains over prior approaches and capabilities. Also creates a new corpora to further the research in the future.

---

### Decision · Program_Chairs · 2024-01-16

Accept (poster)